# Specificity for deubiquitination of monoubiquitinated FANCD2 is driven by the N-terminus of USP1

Connor Arkinson[1], Viduth K Chaugule[1], Rachel Toth[2], Helen Walden[1]

The Fanconi anemia pathway for DNA interstrand crosslink repair and the translesion synthesis pathway for DNA damage tolerance both require cycles of monoubiquitination and deubiquitination. The ubiquitin-specific protease-1 (USP1), in complex with USP1-associated factor 1, regulates multiple DNA repair pathways by deubiquitinating monoubiquitinated Fanconi anemia group D2 protein (FANCD2), Fanconi anemia group I protein (FANCI), and proliferating cell nuclear antigen (PCNA). Loss of USP1 activity gives rise to chromosomal instability. Whereas many USPs hydrolyse ubiquitin–ubiquitin linkages, USP1 targets ubiquitin–substrate conjugates at specific sites. The molecular basis of USP1's specificity for multiple substrates is poorly understood. Here, we reconstitute deubiquitination of purified monoubiquitinated FANCD2, FANCI, and PCNA and show that molecular determinants for substrate deubiquitination by USP1 reside within the highly conserved and extended N-terminus. We found that the N-terminus of USP1 harbours a FANCD2-specific binding sequence required for deubiquitination of K561 on FANCD2. In contrast, the N-terminus is not required for direct PCNA or FANCI deubiquitination. Furthermore, we show that the N-terminus of USP1 is sufficient to engineer specificity in a more promiscuous USP.

## Introduction

Ubiquitination is a reversible post-translational modification that regulates almost every cellular process in eukaryotes. Cycles of ubiquitination and deubiquitination orchestrate the assembly and disassembly of many DNA repair complexes in DNA damage response pathways. These include the Fanconi anemia (FA) pathway, required to repair DNA interstrand crosslinks (ICLs), and the translesion synthesis pathway (TLS), required for DNA damage tolerance (1). FA is a chromosomal instability disorder that results from a dysfunctional ICL repair pathway (2). Central to FA ICL repair is monoubiquitination of two homologous proteins, Fanconi anemia group D2 protein (FANCD2) and Fanconi anemia group I protein

(FANCI), at two specific lysines, K561 and K523, in humans, respectively. Monoubiquitination of FANCD2 and FANCI is catalysed by the E3 ubiquitin ligase FANCL and the E2 conjugating enzyme Ube2T (3). Monoubiquitinated FANCD2 (FANCD2-Ub) signals multiple DNA repair proteins to conduct ICL repair (2). A similar specific modification is central to TLS repair, where K164 of proliferating cell nuclear antigen (PCNA) is monoubiquitinated (PCNA-Ub) by the RAD18 E3 ligase and Rad6 E2 (4), which, in turn, recruits TLS polymerases for DNA repair (5). As well as ubiquitination, both ICL and TLS repair require deubiquitination (removal of ubiquitin). Interestingly, although the modifications in each pathway are assembled by distinct enzymes, they are removed by the same deubiquitinase (DUB), the USP1–USP1-associated factor 1 (UAF1) complex (6, 7, 8). Loss of USP1 function results in an accumulation of FANCD2, FANCI, and PCNA; genomic instability; and a failure to complete the pathways (7, 9, 10, 11, 12). In addition to these three substrates, USP1 deubiquitinates a number of other substrates, including the inhibitor of DNA-binding proteins 1–4 (ID1-4), which regulate cell differentiation (13), and TBK1, which is involved in viral infection (14).

USP1 belongs to the largest family of DUBs, the ubiquitin-specific proteases (USPs), which contain ~50 members. Many USPs show little substrate discrimination between ubiquitin–ubiquitin chains in vitro (15), and can hydrolyse polyubiquitin chains from substrates (16). A few USPs exhibit preference for specific ubiquitin–ubiquitin linkages, such as USP30 for K6-linked Ub chains (17). In contrast, USP1 targets monoubiquitinated substrates and regulates a distinct set of modified proteins. Although molecular mechanisms of ubiquitin removal from ubiquitin are well understood (16), it is less clear how ubiquitin–substrate linkages are specifically targeted. The core catalytic USP domain is ~350 amino acids. However, most USPs also contain multiple insertions within the catalytic domain and additional N/C-terminal extensions (18). USP1 has multiple insertions and an extended N-terminus on its USP domain, and their functions are currently unknown.

USP1 has little DUB activity on its own, but is regulated by and forms a stoichiometric complex with UAF1. UAF1 also binds and activates two other DUBs, USP12 and USP46 (19), and studies that reveal how UAF1 binds and activates USP12 and USP46 suggest that

[1]Institute of Molecular Cell and Systems Biology, University of Glasgow, Glasgow, UK School of Life Sciences, The University of Dundee, Dundee, UK  [2]The Medical Research Council Protein Phosphorylation and Ubiquitylation Unit,

Correspondence: Helen.Walden@glasgow.ac.uk

UAF1 will bind to USP1 in an analogous manner (20, 21). UAF1 acts to stabilise its USP partners and increase catalytic activity (22). UAF1 knockout in mice is embryonic lethal, whereas USP1 knockouts result in a FA-like phenotype, reflecting the additional functions of UAF1 (9,23). In addition to its activation role, UAF1 has a C-terminal SUMO-like domain (SLD) responsible for recruiting USP1 indirectly to FANCD2 and PCNA via a weak interaction with SUMO-like interacting motifs in FANCI and ATAD5, respectively (24). Despite the common activator function of UAF1, loss of either USP12 or USP46 does not result in accumulation of USP1 substrates (19), suggesting that USP1 could target its substrates independent of UAF1. However, it remains unclear how USP1 specifically targets its substrate pool.

Investigating how USPs deubiquitinate their substrates on a molecular level is very challenging because of the difficulty in making physiological and correctly ubiquitinated substrates. To date, most of our understanding of DUB specificity has used ubiquitin–ubiquitin linkages as substrates, likely because of the advances in purifying large quantities of ubiquitin chains. However, there are a few examples of studies that have used monoubiquitinated substrates with a native isopeptide, and these include PCNA-Ub (25) and histones (26). Particularly in the case of histone H2A and H2B, the generation of monoubiquitinated substrates has facilitated the elucidation of the mechanisms of substrate targeting by histone-specific DUBs (26, 27). The ability to make physiological substrates allows for a modular approach to understanding the requirements for specificity and how DUBs such as USP1 work at the molecular level.

Here, we report the reconstitution of substrate deubiquitination by USP–UAF1 that allows for a modular approach to understanding the molecular requirements for deubiquitination of physiological substrates. We define the molecular determinants for substrate deubiquitination via a few residues in a highly conserved and extended N-terminus of USP1. Our analysis indicates that the N-terminus of USP1 harbours a FANCD2-specific binding sequence important for deubiquitinating only one specific lysine on FANCD2: K561—the location of a specific DNA repair signal. Remarkably, we find that the N-terminus of USP1 is important for FANCD2-K561Ub deubiquitination but apparently not for PCNA-K164Ub or FANCI-K523Ub. Finally, we find that the N-terminus of USP1 is sufficient to engineer specificity in a more promiscuous DUB. Our analysis shows that USP1 discriminates between substrates and has direct elements for targeting FANCD2, regardless of whether it is ubiquitinated, providing further insights into how USPs select their substrates.

## Results

### USP1 catalytic domain is sufficient for activity and binding to UAF1

To gain insight into USP1 specificity, we analysed the amino acid sequence and separated the protein into regions termed USP domain, Insert 1, Insert 2, and N-terminus (Fig 1A). Several elements important for cellular regulation have been previously identified, such as a calpain cleavage site within the N-terminus (28), a degron motif for anaphase-promoting complex/cyclosome[Cdh1] targeting within insertion 1 (29), and an auto-cleavage region (G670/G671) within insertion 2 (7). However, whether these regions are important

for in vitro DUB activity is unknown. We designed multiple USP1 fragments for expression in Sf21 insect cells, keeping the catalytic domain intact, but systematically removing the insertions and N-terminus (Fig 1B). Each construct of USP1 is expressed and purified to homogeneity (Fig 1C). To assess catalytic activity and competency, we used *ubiquitin-propargylamine* (Ub-prg), a "suicide" probe which crosslinks to the active site cysteine residue of DUBs (30). Recombinant USP1[FL], in addition to all of the other fragments, fully reacts with Ub-prg (Fig 1C), thus indicating the catalytic domain is competent and able to bind ubiquitin. USP1[FL] has multiple breakdown products, but when insertions 1 and 2 are deleted, the yield and purity are much higher, indicating an increase in protein stability. Indeed, thermal denaturation assays reveal that USP1[ΔNΔ1Δ2] is more thermostable ($T_m$ = 44 ± 0.24°C), whereas USP1[FL] melts at lower temperatures ($T_m$ = 37 ± 0.81°C) (Fig S1). Importantly, USP1 requires to be in complex with UAF1 for robust proteolytic activity (22). We find that UAF1 can bind the minimal USP1 catalytic domain (USP1[ΔNΔ1Δ2]) by size exclusion chromatography (SEC) (Fig 1D) and stimulate DUB activity on a fluorescent ubiquitinated-dipeptide substrate, Ub-KG[TAMRA] (Fig 1E). The cleavage rates show that the activity of a USP1[ΔNΔ1Δ2]–UAF1 complex is almost identical to that of USP1[FL]–UAF1. Because a minimal pseudosubstrate was used, we further assayed the activity against K63- and K48-linked di-ubiquitin chains (Fig 1F) and did not detect any differences in activity between USP1[FL] and USP1[ΔNΔ1Δ2]. Together, these data show that the catalytic domain of USP1, with stoichiometric amounts of UAF1, is sufficient for DUB activity and that the additional regions of USP1 are not important for in vitro DUB catalysis or UAF1 stimulated activity.

### USP1–UAF1 directly deubiquitinates monomeric FANCD2-Ub and FANCI-Ub

The USP1–UAF1 complex can deubiquitinate several monoubiquitinated substrates, including human FANCD2-K561Ub (*hs*FANCD2-Ub) (6), human FANCI-K523Ub (*hs*FANCI-Ub) (31), and human PCNA-K164Ub (PCNA-Ub) (25). To understand the requirements for deubiquitination, we optimised a robust method for producing full-length isolated *hs*FANCD2-Ub and *hs*FANCI-Ub substrates. The homogeneous preparation of isolated *hs*FANCD2-Ub and *hs*FANCI-Ub is very challenging, as previous studies required a non-mammalian FANCD2–FANCI substrate complex, in addition to needing DNA and a six-protein E3 ligase complex (FANCC, FANCE, FANCF, FANCB, FANCL, and FAAP100), to stimulate the modification (32). To enhance the yield and purity of *hs*FANCD2-Ub and *hs*FANCI-Ub, we developed a method that requires only a FANCL fragment (FANCL[ΔELF]) and a hyperactive mutant form of the E2, Ube2Tv4 (33), to stimulate the reaction (see the Materials and Methods section). Using these enzymes (Fig 2A), we observed robust monoubiquitination of the isolated human FANCD2 and FANCI substrates within 60 min when monitored with either fluorescent ubiquitin (Ub[800]) (Fig S2A) or GST-Ub (Fig 2B). To ensure site specificity of ubiquitination, we mutated K561 and K523 to Arg in FANCD2 and FANCI, respectively. In contrast to WT FANCD2 and FANCI, the mutants K561R and K523R are not monoubiquitinated using these reaction conditions, indicating that site specificity is maintained (Fig 2B). We further assessed site specificity by purifying *hs*FANCD2-Ub and *hs*FANCI-Ub to homogeneity and analysed both modified and unmodified proteins

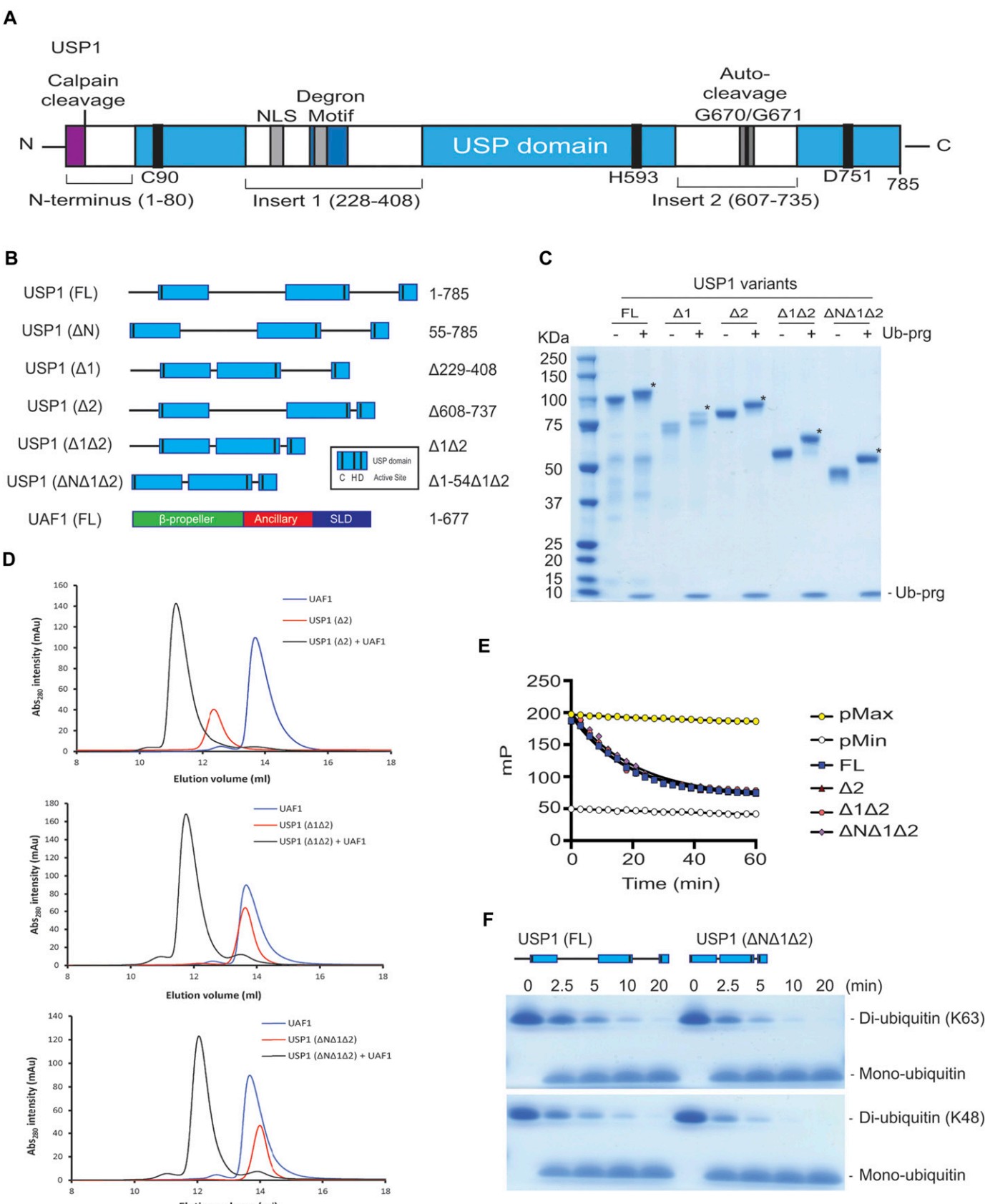

by mass spectrometry to verify whether the correct lysine was ubiquitinated. We detected ubiquitin on FANCD2-K561 and FANCI-K523 and did not detect any secondary ubiquitination events (Fig S2B). In addition, these reaction conditions also support monoubiquitination of the isolated *Xenopus laevis* (*xl*) FANCD2 and FANCI substrates without the need for any reaction cofactors (Fig S2C). Purification profiles of both *hs*FANCD2-Ub and *hs*FANCI-Ub show no apparent changes in their solution states or hydrodynamic radius compared with non-ubiquitinated proteins (Fig 2C). Finally, we purified *hs*PCNA-Ub using a previously described method that uses a mutant E2 (UbcH5c S22R (25)) (Fig 2C). With the substrates purified (Fig 2C), we assessed whether *hs*FANCD2-Ub, *hs*FANCI-Ub, or *hs*PCNA-Ub is directly deubiquitinated by USP1–UAF1 in vitro. The recombinant USP1$^{FL}$ is able to fully deubiquitinate *hs*FANCD2-Ub, *hs*FANCI-Ub, and *hs*PCNA-Ub at sub-stoichiometric concentrations, in a UAF1-dependent manner (Fig 2D). In contrast, a catalytically dead mutant of USP1 (C90S) is unable to deubiquitinate any of the substrates (Fig 2D). Taken together, these data establish that recombinant USP1–UAF1 is able to directly deubiquitinate diverse monoubiquitinated substrates in vitro without the need for auxiliary factors.

### The N-terminus of USP1 is critical for FANCD2-Ub deubiquitination

To understand the minimum requirements for deubiquitination, we assessed whether the catalytic domain of USP1 is sufficient for deubiquitinating its physiological substrates. To test this, we assayed multiple USP1 fragments. As shown, USP1$^{FL}$–UAF1 readily deubiquitinates *hs*FANCD2-Ub, *hs*FANCI-Ub, and *hs*PCNA-Ub (Fig 3A). In addition, the minimal catalytic module (USP1$^{ΔNΔ1Δ2}$–UAF1) is sufficient for deubiquitinating *hs*FANCI-Ub (Fig 3A). Surprisingly, USP1$^{ΔNΔ1Δ2}$–UAF1 is unable to deubiquitinate *hs*FANCD2-Ub and there is slower activity on *hs*PCNA-Ub at early time points (Fig 3A), indicating the catalytic domain is not sufficient despite the catalytic domain being active on other substrates including *hs*FANCI-Ub. In light of this unexpected result, we wanted to determine which deleted region of USP1 is responsible for the loss in deubiquitinating activity on *hs*FANCD2-Ub. We, therefore, assayed each deletion: USP1$^{ΔN}$, USP1$^{Δ1}$, and USP1$^{Δ2}$. Both deletions of inserts 1 and 2 cleave at the same rate as USP1$^{FL}$; however, deletion of the N-terminus results in a large loss of activity on *hs*FANCD2-Ub (Fig 3B). In contrast to *hs*FANCD2-Ub, none of the deletions have any apparent effect on *hs*FANCI-Ub deubiquitination or *hs*PCNA-Ub deubiquitination (Fig 3B). Because the minimal USP1$^{ΔNΔ1Δ2}$ is still an active DUB, we wanted to determine whether at higher concentrations *hs*FANCD2-Ub could be deubiquitinated by this fragment. Full deubiquitination of *hs*FANCD2-Ub can only be achieved by increasing the concentration of USP1$^{ΔNΔ1Δ2}$ to a 1:1 ratio (Fig 3C). This is in contrast to USP1$^{Δ1Δ2}$, which is able to deubiquitinate *hs*FANCD2-Ub at sub-stoichiometric

concentrations (Fig 3C). These data indicate that although the N-terminus is not required for catalytic activity per se, it indeed contributes to a more productive *hs*FANCD2-Ub DUB activity.

A recent report suggests that *xl*FANCI is required for human USP1–UAF1 to remove ubiquitin from *xl*FANCD2-Ub (32). In contrast, we found that *xl*FANCD2 does not require the presence of *xl*FANCI to be deubiquitinated (Fig 3A). Therefore, we speculated whether our observation was because of the use of human FANCD2 substrate. To test this, we purified *xl*FANCD2-Ub to homogeneity (Fig S2C–E) and assayed with human USP1–UAF1. In contrast to previous reports, we found that USP1$^{FL}$ and USP1$^{Δ1Δ2}$ are able to fully deubiquitinate *xl*FANCD2-Ub (Fig 3D). Furthermore, consistent with our observations using *hs*FANCD2-Ub as substrate, deletion of the N-terminus reduces USP1 activity for *xl*FANCD2-Ub (Fig 3D). Because deletion of the N-terminus of USP1 does not result in an apparent defect for Ub-KG$^{TAMRA}$ (Fig 1E), K63/K48 diUb (Fig 1F), *hs*FANCI-Ub (Fig 3B), and *hs*PCNA-Ub (Fig 3B), it is likely that the USP1 N-terminus is a specific FANCD2-Ub requirement that is extended from the catalytic domain. Taken together, these data suggest that the N-terminus of USP1 harbours a substrate targeting sequence specific for FANCD2.

### The N-terminus drives specific FANCD2-K561-Ub deubiquitination

The ubiquitination site of FANCD2 is at a specific residue, K561. We speculated whether the FANCD2 targeting by the N-terminus of USP1 also extends to the site of ubiquitination, that is, for K561-Ub. To test this, we generated FANCD2 and modified lysines distinct from K561 (Fig 4A). We decided to use a mutant E2 (UbcH5c$^{S22R}$) that primarily monoubiquitinates proteins (25). As the monoubiquitination activity was weak, we used the E3 ligase RNF4 fragment (RNF4-RING fusion, RNF4$^{RR}$) to increase the activity (34). The UbcH5c$^{S22R}$/RNF4$^{RR}$ pair resulted in robust and multiple monoubiquitination events on *hs*FANCD2 (Fig 4B). To ensure K561 is not being monoubiquitinated by UbcH5c$^{S22R}$/RNF4$^{RR}$, we used K561R *hs*FANCD2 as the substrate. We performed an E3 assay to monoubiquitinate *hs*FANCD2 and *hs*FANCD2 K561R, which we term *hs*FANCD2 K561-Ub and *hs*FANCD2 KX-Ub, respectively (Fig 4A), and treated the reaction products with USP1$^{FL}$–UAF1 or USP1$^{ΔN}$–UAF1. Interestingly, whereas USP1$^{FL}$–UAF1 deubiquitinates both substrates K561-Ub and KX-Ub at similar rates, USP1$^{ΔN}$–UAF1 shows clear activity on KX-Ub but little detectable activity on K561-Ub (Fig 4C). These data show that the USP1 N-terminus targeting sequence also specifies the site for DUB activity on FANCD2-Ub, that is, K561-Ub.

### Comparing deubiquitination activity of *E. coli*– and Sf21-expressed USP1

To determine whether the N-terminus dependence of *hs*FANCD2-Ub deubiquitination by USP1 was due to possible post-translational

**Figure 1. The catalytic domain of USP1 is sufficient for UAF1 binding and activation.**
**(A)** Overall schematic of USP1 domain structure and boundaries. USP1 contains an extended N-terminus and two large insertions within the USP domain. This schematic excludes minor insertions within the catalytic domain. Included: Degron motif (295–342) for APC/C$^{Cdh1}$, NLS (266–287 and 298–321), catalytic triad residues (C90, H592, and D751), and a calpain cleavage site (S13). **(B)** Schematic of truncation/deletion fragments of USP1 and UAF1 used in this study. **(C)** Recombinant USP1 constructs from Sf21 cells incubated with threefold molar excess ubiquitin-propargylamine (Ub-prg). A band shift (*) shows crosslinking of Ub-prg to USP1. **(D)** SEC of recombinant USP1 with UAF1 at 15 μM. Catalytic domain of USP1 (USP1$^{ΔNΔ1Δ2}$) is sufficient for forming a complex with UAF1. **(E)** A comparison of DUB activity of USP1 constructs using 300 nM Ub-KG$^{TAMRA}$ as a minimal substrate. Equimolar UAF1 was used as USP1 alone contains little activity. **(F)** Hydrolysis of K63- or K48-linked di-ubiquitin (10 μM) using 50 nM USP1$^{FL}$–UAF1 or 50 nM USP1$^{ΔNΔ1Δ2}$–UAF1. Data show comparable activity of USP1$^{FL}$ and the catalytic domain of USP1. FL, full length; NLS, nuclear localisation signals. Source data are available online for this figure.

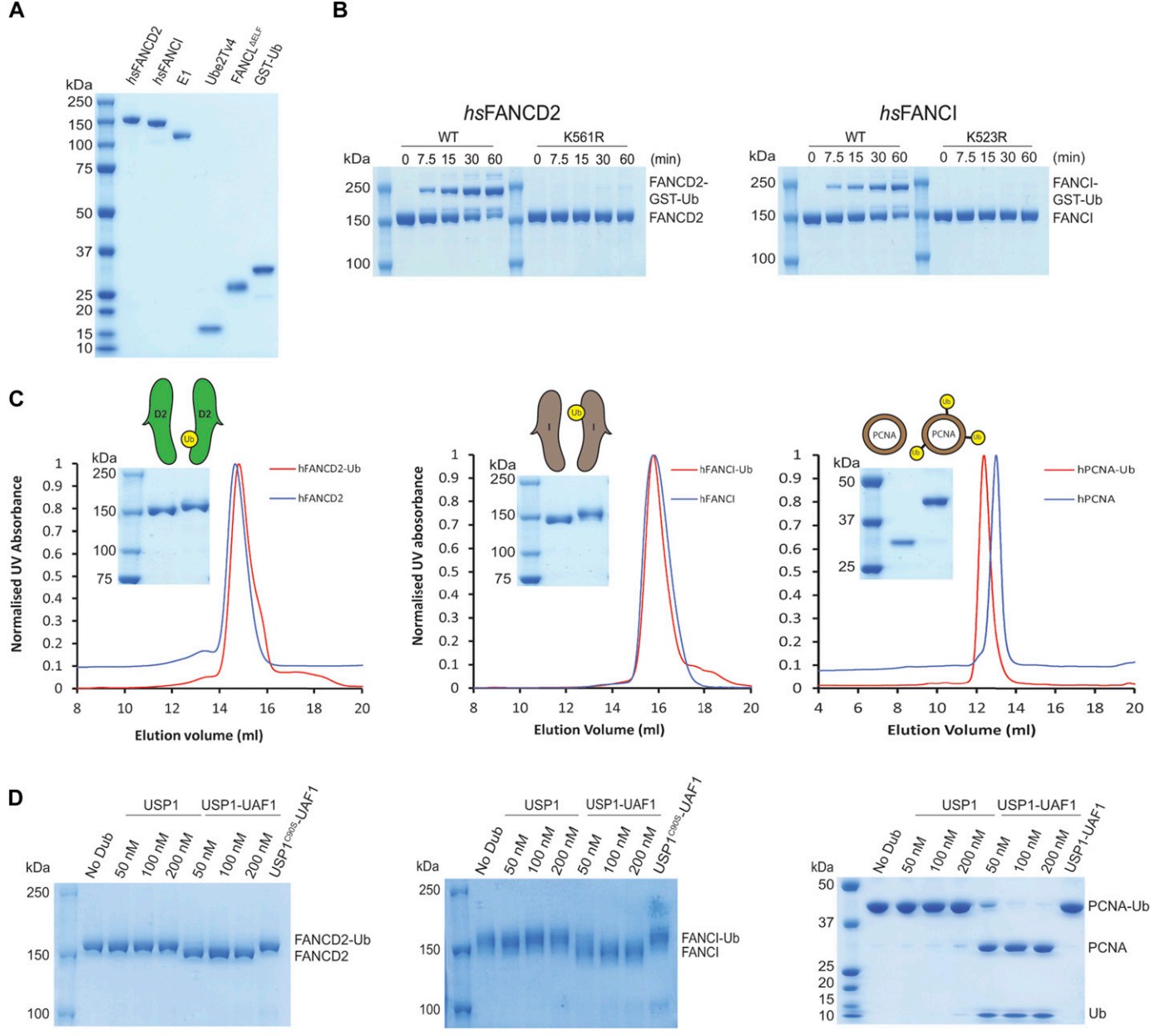

**Figure 2. Recombinant human FANCD2-Ub and FANCI-Ub are deubiquitinated by USP1–UAF1 in vitro.**
**(A)** Purification of human FANCD2 and FANCI, a hyperactive mutant Ube2Tv4, FANCL^ΔELF, ubiquitin E1, and GST-ubiquitin. **(B)** Ubiquitination reactions using reaction components in (A). FANCD2 with K561R and FANCI with K523R mutations were used as a control for lysine specificity. **(C)** Purity of monoubiquitinated human *hs*FANCD2 and *hs*FANCI, and SEC comparing modified and unmodified *hs*FANCD2 and *hs*FANCI. **(D)** Deubiquitination (40 min) of 1 μM *hs*FANCD2-Ub, *hs*FANCI-Ub, and *hs*PCNA-Ub by 100 nM USP1 with and without UAF1.
Source data are available online for this figure.

modifications in eukaryotic cells, we also expressed and purified USP1^Δ1Δ2 and USP1^ΔNΔ1Δ2 from *E. coli* and tested their activity (Fig S3). A comparison of Sf21- and *E. coli*–expressed USP1 shows only minor differences in activity on *hs*FANCD2-Ub, and confirms that the activity depends on the N-terminus of USP1 (Fig S3A). Interestingly, a small difference in activity is apparent for *hs*PCNA-Ub deubiquitination when comparing USP1^Δ1Δ2 and USP1^ΔNΔ1Δ2 (Fig S3B), perhaps indicating some form of regulation within the N-terminus for targeting *hs*PCNA.

However, as deubiquitination of *hs*FANCD2-Ub still depends on the N-terminus regardless of how USP1 is expressed, we used the *E. coli* USP1 to assess point mutations within the N-terminus.

### The N-terminus of USP1 contains critical residues for FANCD2-Ub deubiquitination

To identify residues within the N-terminus of USP1 required for *hs*FANCD2-Ub specificity, we looked at regions that are well conserved

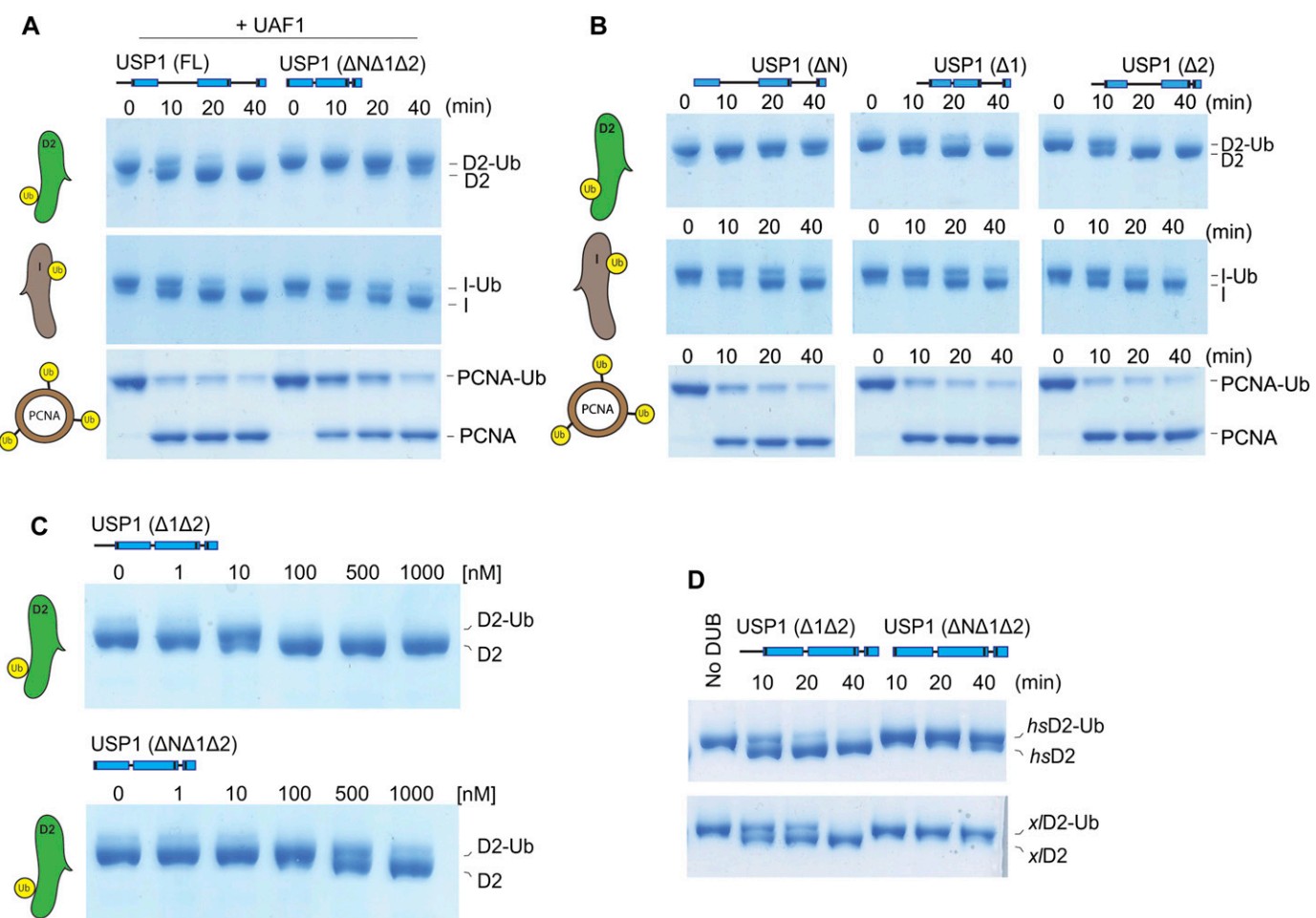

**Figure 3. The N-terminus of USP1 is required for robust FANCD2-Ub deubiquitination but not for hFANCI-Ub or PCNA-Ub.**
**(A)** Deubiquitination reactions of 1 μM recombinant *hs*FANCD2-Ub, *hs*FANCI-Ub, and 3 μM *hs*PCNA-Ub using 100 nM of either USP1$^{FL}$–UAF1 or USP1$^{ΔNΔ1Δ2}$–UAF1. **(B)** Deubiquitination assays of *hs*FANCD2-Ub, *hs*FANCI-Ub, and *hs*PCNA-Ub using indicated USP1 truncations. **(C)** Full deubiquitination of 1 μM *hs*FANCD2 in 30 min requires 10× USP1$^{ΔNΔ1Δ2}$ when compared to USP1$^{Δ1Δ2}$. **(D)** Deubiquitination of frog (*X. laevis*) FANCD2 by USP1$^{Δ1Δ2}$ and USP1$^{ΔNΔ1Δ2}$. Proteins are separated by SDS–PAGE and stained with Coomassie blue.
Source data are available online for this figure.

within the N-terminus of USP1 (Fig 5A). The N-terminus of USP1 is almost completely conserved from humans to zebrafish between residues 19 and 40 (Fig 5A). Similar to deleting residues 1–54 of USP1, deletion of residues 21–29 also results in a specific loss in activity for *hs*FANCD2-Ub but not *hs*PCNA-Ub (Fig 5B). We next determined whether any of the side chains are important, so we mutated each residue to alanine and assayed for DUB activity using Ub-prg, *hs*PCNA-Ub, *hs*FANCI-Ub, and *hs*FANCD2-Ub (Fig S4). All point mutants are active with Ub-prg (Fig S4A), and deubiquitinate *hs*PCNA-Ub and *hs*FANCI-Ub (Fig S4B). In contrast, there is a loss in activity for several point mutants, such as N21A, R22A, L23A, S24A, and K26A, on *hs*FANCD2-Ub deubiquitination (Fig S4B). To quantify the relative differences between USP1 mutants, we purified *hs*FANCD2-Ub with a fluorescently labelled Ub$^{800}$ that allows us to detect only the ubiquitinated FANCD2 (*hs*FANCD2-Ub$^{800}$) (Fig S5). Using this approach, we monitored the deubiquitination of *hs*FANCD2-Ub$^{800}$ by each mutant and quantified the remaining FANCD2-Ub as a percentage of the input (Fig 5C). We found that whereas USP1$^{Δ1Δ2}$–UAF1

cleaves ~80% of the *hs*FANCD2-Ub$^{800}$, USP1$^{ΔNΔ1Δ2}$–UAF1 shows no apparent cleavage (Fig 5C). Interestingly, R22A and L23A are able to only deubiquitinate ~10% of the *hs*FANCD2-Ub$^{800}$ (Fig 5C). We did not further characterise L23A as it co-purifies with a large contaminant chaperone, which may suggest some misfolding (Fig S4). We next assessed whether the charge at R22 is critical for deubiquitination as it is conserved as a lysine in zebrafish (Fig 5A). We generated R22K, R22A, and R22E variants (Fig S4C), and as expected, these had no apparent effect on *hs*PCNA-Ub or *hs*FANCI-Ub deubiquitination (Figs 5D and S4D). In contrast, we observed that R22A has an intermediary affect and R22E is comparatively more compromised on *hs*FANCD2-Ub deubiquitination (Fig 5D). R22K retains WT activity on *hs*FANCD2-Ub (Fig 5D), suggesting a positive charge is required. Finally, we found that in the full-length USP1–UAF1 context, a single point mutation in its N-terminus (R22E) is sufficient for specifically disrupting *hs*FANCD2-Ub deubiquitination but not *hs*FANCI-Ub or *hs*PCNA-Ub (Fig S6A–C). Taken together, these data indicate that R22 plays an important role in deubiquitination of

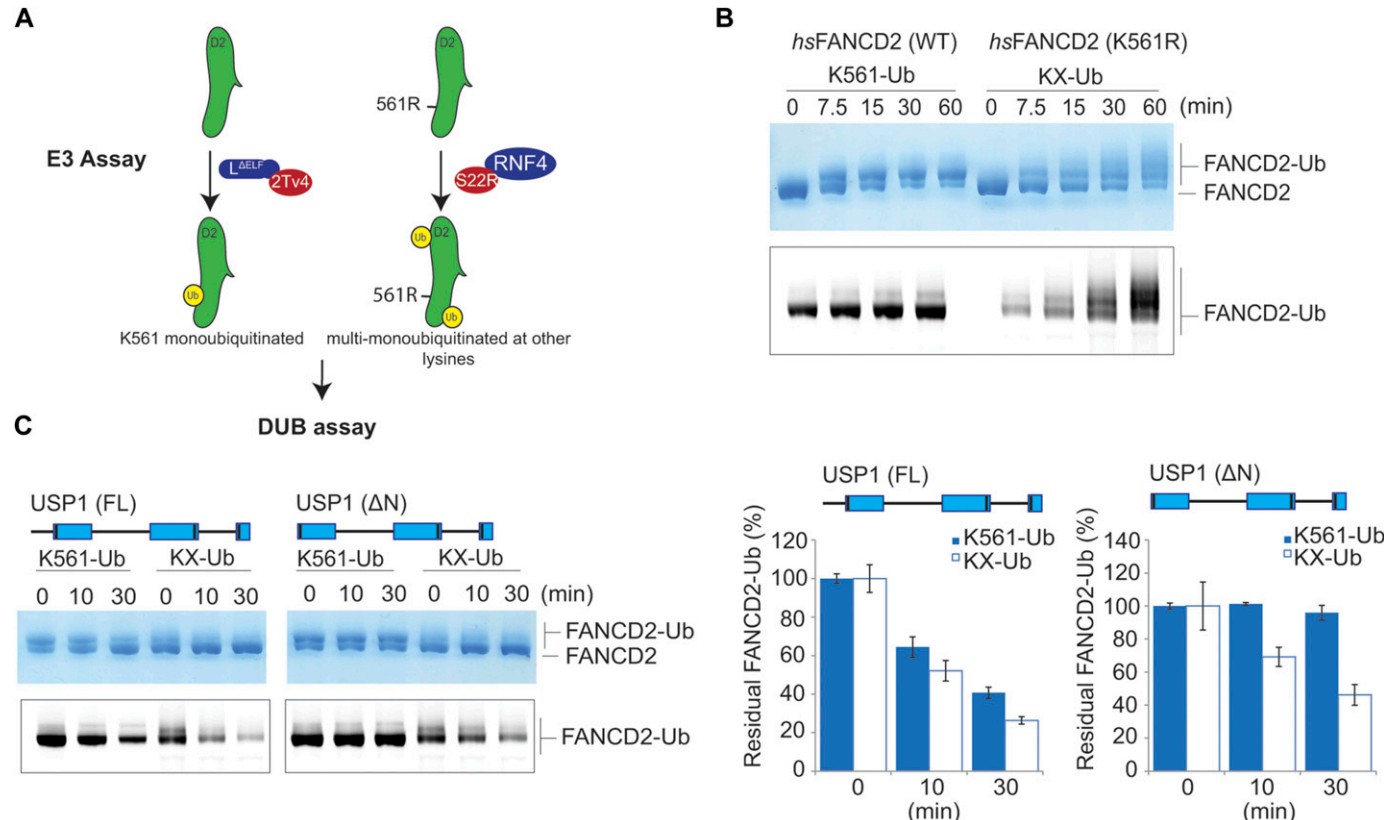

**Figure 4. The N-terminus of USP1 drives specifc K561 deubiquitination of FANCD2.**
**(A)** Schematic of *hs*FANCD2 monoubiquitination reactions to make monoubiquitinated *hs*FANCD2-K561-Ub and *hs*FANCD2-KX-Ub (where X denotes random lysines). **(B)** Monoubiquitination of *hs*FANCD2 (WT) by Ube2Tv4 and FANCL$^{ΔELF}$ is predominantly a single monoubiquitination event at K561. Monoubiquitination of *hs*FANCD2 K561R by UbcH5c$^{S22R}$ and RNF4$^{RR}$ (RNF4-RING fusion) results in multi-monoubiquitination of other FANCD2 lysines. Ubiquitinated products are formed in a time-dependent manner as shown by Coomassie staining and fluorescently labelled ubiquitin, which only shows ubiquitinated products. **(C)** Monoubiquitinated FANCD2-K561-Ub or FANCD2-KX-Ub reactions were arrested by treating with apyrase and subjected to deubiquitination by either USP1$^{FL}$–UAF1 or USP1$^{ΔN}$–UAF1. Fluorescent ubiquitin is used to monitor ubiquitinated species and quantify DUB activity. Data show intensities as percent of ubiquitinated input (n = 3) ± SD.
Source data are available online for this figure.

*hs*FANCD2-Ub. However, it is clear that multiple conserved residues (N21, R22, L23, S24, and K26) that are within a short N-terminal region may play a role in driving USP1-mediated FANCD2-Ub deubiquitination.

Mutation or deletion of the USP1 N-terminus specifically and negatively impacts FANCD2-Ub deubiquitination. We hypothesised that this region in the N-terminus of USP1 encodes a FANCD2-binding site. To test this, we purified UAF1 with an N-terminal Strep tag and formed stable complexes with USP1$^{Δ1Δ2}$, R22A, R22K, or USP1$^{ΔNΔ1Δ2}$. We assayed interaction with *hs*FANCD2 and found that a USP1$^{Δ1Δ2}$–UAF1 complex can capture the non-ubiquitinated substrate (Fig S7A). In contrast, when using USP1$^{ΔNΔ1Δ2}$–UAF1 as bait, *hs*FANCD2 is not enriched above the levels seen in the beads control (Fig S7A). These data indicate the USP1 binding to *hs*FANCD2, which is almost completely dependent on the N-terminus of USP1. In addition, USP1$^{Δ1Δ2}$-R22A only partially interacts with *hs*FANCD2, and USP1$^{Δ1Δ2}$-R22K does not weaken the substrate interaction—both results are consistent with the *hs*FANCD2-Ub deubiquitination assays (Figs S7A and 5A). As USP1's interaction with FANCD2 is dependent on the N-terminus, we wanted to determine whether this is specific for *hs*FANCD2, so we similarly assayed interactions of *hs*FANCI and *hs*PCNA. Interestingly, we can detect a weak interaction with both

*hs*FANCI and *hs*FANCI-Ub that is unaffected by deletion of the N-terminus (Fig S7B). We could not detect an interaction between USP1 and *hs*PCNA, but we observed an interaction with *hs*PCNA-Ub (Fig S7C), which is, however, not dependent on the N-terminus of USP1. Together, these data indicate that the N-terminus of USP1 directly interacts with FANCD2.

### A chimera fusion of USP1 N-terminus to the USP2 catalytic domain provides FANCD2 specificity

Deletion of the N-terminus of USP1 does not appear to affect the catalytic activity of the USP1 catalytic module (USP1$^{ΔNΔ1Δ2}$), except in case of FANCD2-Ub deubiquitination. We, therefore, speculated whether the N-terminus can influence the efficiency of a distinct DUB. To test this, we chose the catalytic domain of USP2, generally considered a versatile/promiscuous DUB because of its lack of substrate specificity (35). We created and purified a USP2 chimera (USP1$^{1–60}$-USP2) where residues 1–60 of USP1 are present in the N-terminal to the USP2 catalytic domain. From the in vitro assays, we found that whereas 100 nM of USP2 is able to completely deubiquitinate 1 *µ*M of *hs*FANCD2-Ub by 40 min (Fig 6A), the

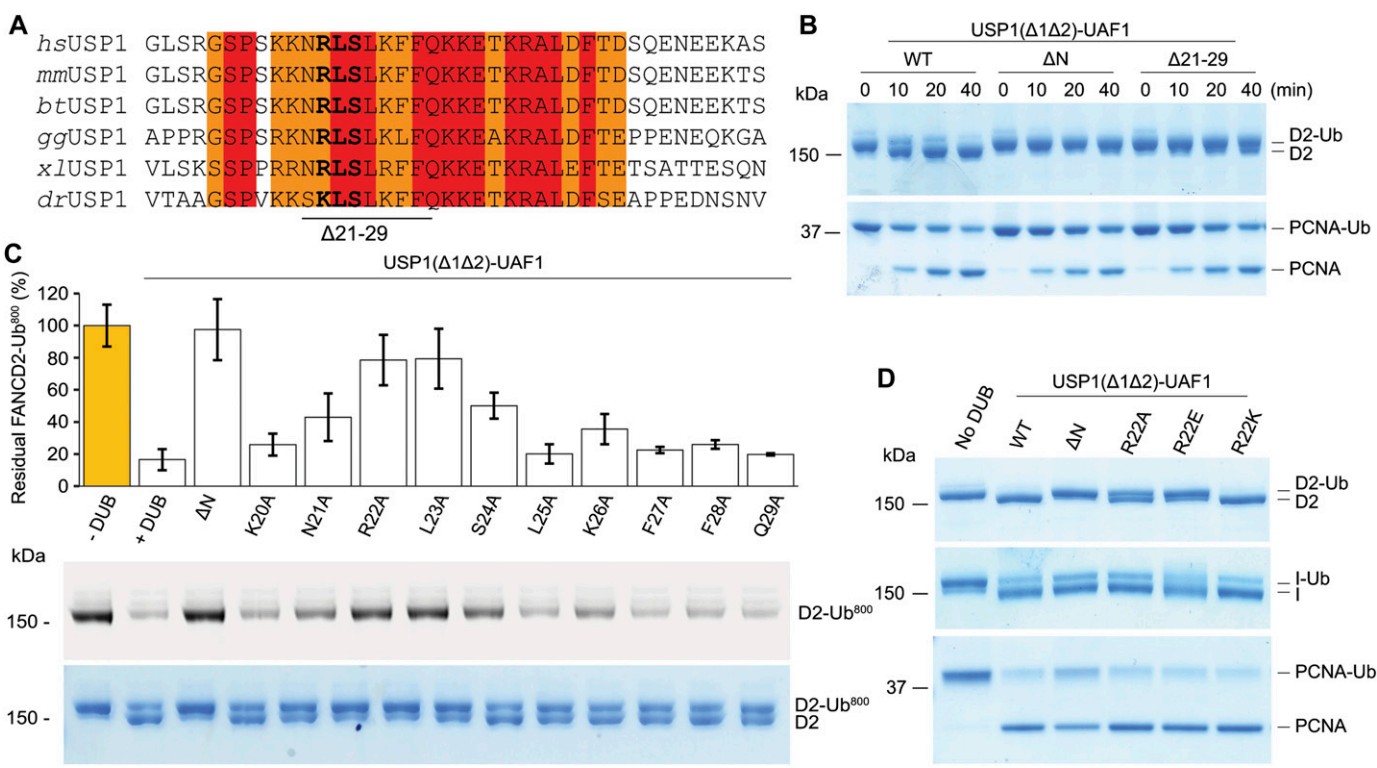

**Figure 5. Several highly conserved residues within USP1 N-terminus determine FANCD2-Ub deubiquitination.**
**(A)** Sequence alignment of USP1 N-terminus from several species (hs, *Homo sapiens*; mm, *Mus mucus*; bt, *Bos taurus*; gg, *Gallus gallus*; xl: *X. laevis*; dr, *Danio rerio*). Deleted USP1 residues in (B) and residues important for acitivity (bold) in (C) are indicated. **(B)** Deubiquitination assay of *hs*FANCD2-Ub and *hs*PCNA-Ub using *E. coli*–purified USP1$^{\Delta1\Delta2}$, USP1$^{\Delta N\Delta1\Delta2}$, and USP1$^{\Delta21-29\Delta1\Delta2}$ in complex with UAF1. **(C)** Deubiquitination assays of *hs*FANCD2-Ub$^{800}$ by recombinant USP1$^{\Delta1\Delta2}$ with Ala point mutants from *E. coli*. The loss in flurorescence intensity was quantified and normalised to the amount of input *hs*FANCD2-Ub$^{800}$. Data display mean residual FANCD2-Ub$^{800}$ (%) (n = 3) ± SD normalised to input. **(D)** Deubiquitination assays of *hs*FANCD2-Ub for 40 min using USP1$^{\Delta1\Delta2}$ with R22 point mutations indicate that a positive charge at position 22 is critical for *hs*FANCD2-Ub deubiquitination but not for *hs*FANCI-Ub or *hs*PCNA-Ub.
Source data are available online for this figure.

chimeric USP1$^{1–60}$-USP2 achieves this within 10 min (Fig 6A), showing a clear gain of activity. Because the N-terminus increases USP2 activity against FANCD2-Ub, we tested whether the N-terminus augments its activity towards other ubiquitinated substrates. In contrast to *hs*FANCD2-Ub, there are no obvious differences in the activity of USP2 and the USP2 chimera using *hs*FANCI-Ub as the substrate (Fig 6B). In addition, only minor differences in the deubiquitination activity of *hs*PCNA-Ub are observed at early time points (Fig 6C). These data show that the addition of USP1 N-terminus in *cis* confers a gain of activity for USP2 on *hs*FANCD2-Ub.

### FANCD2–FANCI–DNA complex deubiquitination depends on the N-terminus of USP1

The FANCD2 and FANCI substrates can form a heterodimer on DNA, and this ensemble has been shown to be important for in vitro monoubiquitination (32, 36, 37). However, it is not yet known how the monoubiquitinated FANCD2 and FANCI proteins assemble or interact once modified. It is, therefore, unclear whether *hs*FANCD2-Ub is deubiquitinated as part of a complex or on its own. A recent report has showed that the di-monoubiquitinated frog ID2 complex (both FANCD2 and FANCI monoubiquitinated), generated in the presence of plasmid DNA, remains resistant to deubiquitination by

a GST-tagged USP1–UAF1 complex and only the removal of DNA permits efficient DUB activity (32). Because under physiological conditions the monoubiquitinated ID2 complex is normally localized on DNA, we therefore wanted to test whether deubiquitination of the *hs*ID2–DNA complex is also dependent on the N-terminus of USP1. To test this, we first monoubiquitinated the *hs*ID2 heterodimer in the presence of double stranded DNA (dsDNA) (36) and arrested the ubiquitination reactions with apyrase. We subsequently treated half of the substrate sample with benzonase nuclease to eliminate the DNA before subjecting both the nuclease-treated and untreated substrate samples to USP1–UAF1 activity. Whereas USP1$^{FL}$ is able to deubiquitinate some of the *hs*FANCD2-Ub in the presence of both DNA and *hs*FANCI, a portion of *hs*FANCD2-Ub and *hs*FANCI-Ub remains resistant to deubiquitination (Fig 7A), consistent with the recent findings using frog substrates (32). Interestingly, when DNA is removed, all of the *hs*FANCD2-Ub and *hs*FANCI-Ub is deubiquitinated by USP1$^{FL}$–UAF1; in contrast, USP1$^{\Delta N}$–UAF1 and USP1$^{R22E}$–UAF1 do not deubiquitinate *hs*FANCD2-Ub (Fig 7A). To allow a clear assessment of whether DNA, FANCI-Ub, or both factors inhibit deubiquitination, we monoubiquitinated FANCD2–FANCI$^{K523R}$–DNA complexes. Using this set-up, we observed that USP1$^{FL}$–UAF1 is able to deubiquitinate *hs*FANCD2-Ub-FANCI with and without benzonase treatment (Fig 7B). These data suggest that

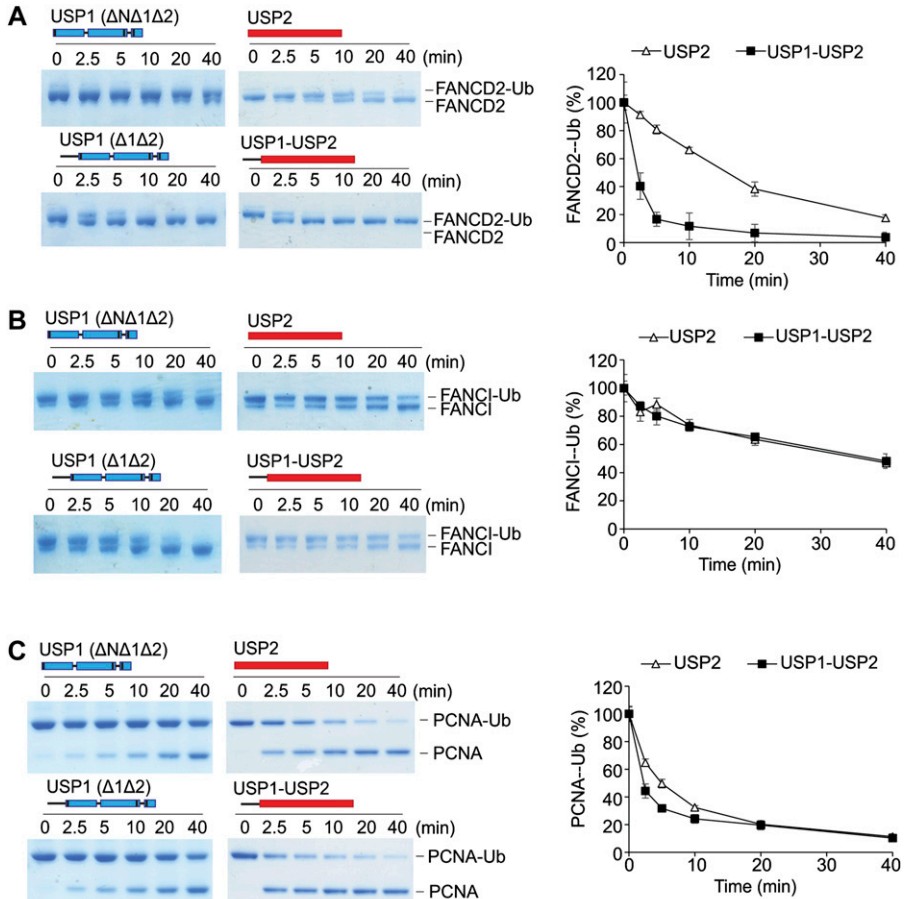

**Figure 6. The N-terminus provides a gain of FANCD2-Ub activity when fused to USP2.**
**(A)** Deubiquitination of 1 µM FANCD2-Ub by either 100 nM of E. coli–expressed USP1$^{\Delta 1 \Delta 2}$–UAF1, USP1$^{\Delta N \Delta 1 \Delta 2}$–UAF1, rat USP2 catalytic domain (USP2cat), or USP1(1–60)-USP2cat chimera, where residues 1–60 of USP1 are fused to the N-terminus of USP2's catalytic domain. Coomassie-stained FANCD2-Ub was scanned and intensity of the FANCD2-Ub band measured over time. Data show mean residual FANCD2-Ub signal (% of input) plotted against time (n = 3) ± SD. **(B)** 1 µM FANCI-Ub and **(C)** 3 µM PCNA-Ub deubiquitination as above. Data show mean residual FANCI-Ub or PCNA-Ub signal (% of input) plotted against time (n = 3) ± SD. Source data are available online for this figure.

DNA does not inhibit FANCD2-Ub deubiquitination except in the context of the di-monoubiquitinated ID2–DNA complex. Interestingly, and despite the presence of FANCI, which may also recruit the DUB complex (24), USP1$^{\Delta N}$–UAF1 and USP1$^{R22E}$–UAF1 are not able to fully deubiquitinate hsFANCD2-Ub, suggesting that any indirect recruitment of USP1–UAF1 by FANCI does not completely compensate for the loss of the USP1 N-terminus. Thus, the N-terminus of USP1 contains a FANCD2 determinant sequence that functions at the core of FANCD2

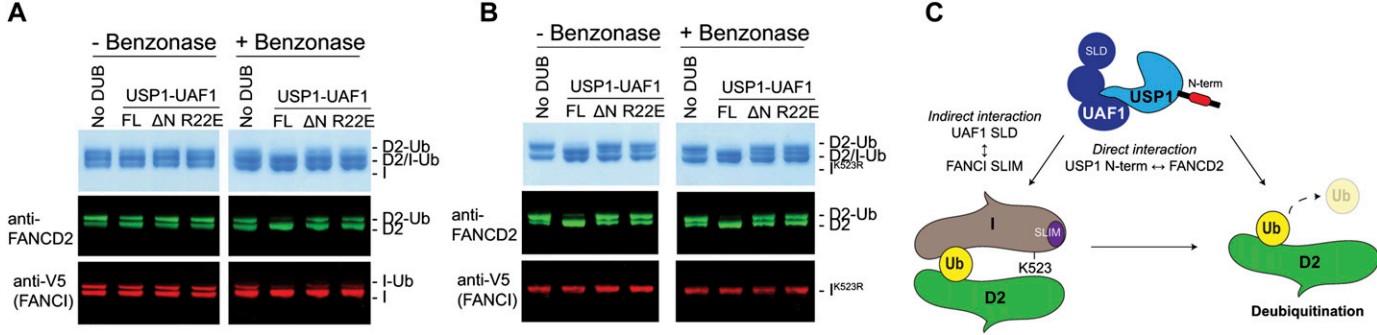

**Figure 7. The N-terminus of USP1 is required for FANCD2-Ub deubiquitination when in complex with FANCI and DNA.**
**(A)** Deubiquitination of di-monoubiquitinated FANCD2–FANCI complex with and without DNA by USP1$^{FL}$–UAF1, USP1$^{\Delta N}$–UAF1, and USP1$^{R22E}$–UAF1. A FANCD2–FANCI–ds DNA complex is monoubiquitinated by Ube2Tv4, FANCL$^{\Delta ELF}$, E1, and ubiquitin for 30 min and reactions arrested using apyrase. Ubiquitination reactions are treated with benzonase nuclease for 4 h on ice before being subject to USP1 for 30 min at room temperature. FANCD2 and FANCI are detected by Coomassie staining or Western blotting on the same nitrocellulose membrane and visualised using anti-FANCD2 or anti-V5 (FANCI). **(B)** Assay performed as in (A) except that FANCI with a K523R mutation is used. **(C)** Model of FANCD2-Ub deubiquitination by USP1–UAF1. USP1 targets FANCD2 via a direct interaction using its N-terminus. In addition to the N-terminus of USP1, FANCD2-Ub–FANCI complexes might be indirectly targeted by an interaction between UAF1 SLD and FANCI SUMO-like interacting motif (SLIM). Source data are available online for this figure.

deubiquitination by specifically binding and increasing deubiquitination efficiency for FANCD2-Ub even in the presence of both dsDNA and FANCI (Fig 7C).

## Discussion

A cycle of monoubiquitination and deubiquitination of FANCD2 is critical for completion of FA ICL repair (8). Removal of the ubiquitin signal from FANCD2 is conducted by the USP1–UAF1 DUB, and disruption of USP1 catalytic activity results in an accumulation of FANCD2-Ub and FA-like phenotypes (6, 9, 11, 23). Despite multiple DUBs existing at DNA replication and repair events, USP1 appears to be the only DUB whose loss results in an accumulation of FANCD2-Ub (6). The molecular determinants and mechanisms of how DUBs, such as USP1, target and regulate distinct pools of substrates have remained elusive, because of a lack of molecular insights. Here, we report a unique feature and a potential short linear motif at the N-terminus of USP1, which shows that USP1 specifically and directly targets K561 of FANCD2, providing a molecular foundation for how USP1 targets one of its substrates with high specificity.

USP1 activity is subject to layers of regulation during cell-cycle progression. For example, USP1 is turned over throughout the cell cycle, is activated and stabilised by binding partners such as UAF1, and, in addition, undergoes autocleavage to allow targeting by both the N-end rule and C-end rule pathways (7, 11, 22, 38). Interestingly, UAF1 is an abundant protein with multiple functions, including activating USP12 and USP46 (19). It has been suggested that UAF1 may indirectly target USP1 to specific substrates such as FANCD2 or PCNA via a C-terminal SLD that binds to SUMO-like interacting motifs on protein partners of FANCD2 and PCNA (24) (Fig 7C). However, disruption of either USP12 or USP46 does not affect FANCD2-Ub levels, showing that FANCD2 is not an overlapping substrate (19). Because the SLD of UAF1 would target multiple DUBs to the same substrates, further layers of regulation may exist to ensure the correct DUB is targeted to FANCD2-Ub or PCNA-Ub. By reconstituting FANCD2-Ub, FANCI-Ub, and PCNA-Ub deubiquitination by USP1–UAF1, we identified the minimal components required for deubiquitination and discovered a unique sequence within the N-terminus of USP1 critical for removing ubiquitin from FANCD2 (Fig 7C). We noticed a relatively small, but reproducible difference in activity on PCNA-Ub, which is apparent when deleting the N-terminus. Dependence on the N-terminus for PCNA-Ub deubiquitination may become more apparent when PCNA-Ub is within physiological contexts, such as being DNA bound and interacting with proteins such as polη (5). Further studies may reveal additional layers of specificity for substrates such as PCNA-Ub, which could also be mediated by the N-terminus of USP1–UAF1. Moreover, it remains unclear whether the insertions of USP1 play a more direct role in substrate specificity and deubiquitination.

The N-terminus of USP1 binds specifically to FANCD2, and not FANCI or PCNA. Thus, USP1 appears to display a higher level of substrate targeting for FANCD2. It is interesting that the N-terminus of USP1 is not required for FANCI-Ub deubiquitination in vitro because FANCI shares a similar ubiquitination site with FANCD2 both structurally (39) and on a sequence level, allowing us to

speculate that perhaps FANCD2 is the major substrate of USP1. Although we showed that FANCD2-Ub with FANCI and DNA does not obviously affect USP1–UAF1 deubiquitination in vitro, FANCI has been shown to regulate FANCD2 deubiquitination in cells via phosphomimetic mutations (40) and SUMO-like interacting motif mutations (24). Therefore, under certain conditions, FANCI may also play a role in the direct regulation of FANCD2-Ub deubiquitination. For example, we and others have shown that di-monoubiquitinated DNA-bound ID2 complexes remain resistant to deubiquitination (32), suggesting that FANCD2-Ub and FANCI-Ub in the presence of DNA reciprocally regulate their deubiquitination by USP1–UAF1. How di-monoubiquitination of ID2–DNA regulates the ability of USP1–UAF1 to remove ubiquitin from FANCD2 will be an interesting mechanism to investigate. For example, does the ID2–DNA complex affect the USP1 N-terminus binding or directly occlude the ubiquitin tail from proteolytic cleavage?

The N-terminus of USP1 is an extension of the USP domain and is ~75 amino acids in length. It is unclear how flexible, structured, or compact the N-terminus will be in a relative position to the USP domain and catalytic site. Interestingly, the N-terminus is modular, as it can be added to other DUB catalytic domains such as USP2 and enhance FANCD2-Ub–specific deubiquitination. Therefore, it is difficult to predict whether the N-terminus of USP1 directly contributes to catalysis when FANCD2 is bound, or simply acts to recruit FANCD2-Ub and stabilise the isopeptide bond within the active site. Because USP1$^{\Delta N}$–UAF1 loses activity for ubiquitinated K561, but not for other lysines of FANCD2, this allows us to speculate that the N-terminus binds and precisely positions FANCD2 to increase activity for a specific lysine position. Another possibility is that FANCD2-K561-Ub is not in a fully accessible conformation for deubiquitination and that the N-terminus modulates this conformation to allow the formation of a more productive complex. This is reminiscent of a recently identified monoubiquitination site of SETDB1, which protects the ubiquitin from deubiquitination via multiple SETDB1–ubiquitin interactions (41). Perhaps FANCD2 also partially protects and occludes ubiquitin at K561, and the N-terminus of USP1 is required to relieve this effect. It will be interesting to determine how the N-terminus binds to FANCD2, how it modulates USP1 function towards FANCD2, and whether other PTMs or processes within the N-terminus can modulate its function.

It is not understood how the majority of USPs target their substrates specifically. Some USPs target the signal itself; for example, USP18 targets ISG15 (42) and USP30 targets K6-linked ubiquitin chains (17, 43). However, here we have an example: USP1, which we have shown targets a specific monoubiquitinated substrate and, therefore, recognises both ubiquitin and the substrate itself. Another specific monoubiquitinated substrate is histone 2B, which is monoubiquitinated on K120 and targeted by the Spt-Ada-Gcn5 acetyltransferase (SAGA) DUB module, which contains USP22 (UBP8 homologue). Whereas UBP8 has contributions for targeting nucleosomes, other members of the complex such as sgf11 make more significant contributions via the acidic patch on H2A/H2B, and therefore likely govern most of the substrate specificity (27). Whereas SAGA DUB achieves specificity via a protein complex, other DUBs such as USP7 contain substrate-targeting regions within the same polypeptide. USP7 displays substrate-targeting regions via its N-terminal tumor necrosis factor–receptor associated factor-like

domain for p53 (44). However, in contrast to the effect we see with the disruption of USP1's N-terminus on FANCD2-Ub deubiquitination, disruption of the N-terminus of USP7 has only a minor effect on p53 deubiquitination in vitro (45). We show that USP1 specifically targets one ubiquitinated lysine made by a specific E3 ligase, FANCL. Another DUB, USP48, was recently published to also specifically oppose the monoubiquitination made by another lysine-specific E3 ligase, BRCA1 (26). This suggests that some DUBs and E3 activities may have co-evolved to oppose each other on specific cellular signals and maintain an appropriate equilibrium of non-ubiquitinated and ubiquitinated substrates.

Arguably, the most important known function of USP1 is to regulate FANCD2, as the monoubiquitinated form is important for both ICL repair and origin of replication firing in latent conditions (46). In addition, non-ubiquitinated FANCD2 is important in DNA surveillance mechanisms (47), meaning the presence of both FANCD2-Ub and FANCD2 is important for DNA integrity and cell proliferation. Moreover, USP1 depletion increases oncogene-induced senescence and plays a pivotal role in protecting genomic instability by preventing FANCD2-Ub aberrant aggregation (12). USP1–UAF1, therefore, works to recycle and maintain appropriate levels of both monoubiquitinated and non-ubiquitinated FANCD2. Inhibition of USP1–UAF1 activity by the specific small molecule ML323 leads to an increased sensitivity to platinum-based compounds in resistant cells (11). Furthermore, the USP1–UAF1 complex has been shown to play critical roles in homologous recombination (10). This collection of studies reveals that USP1 would be an excellent anticancer target by disrupting its ability to deubiquitinate FANCD2-Ub, as this would lead to oncogene-induced senescence and cellular sensitivity to crosslinking agents (8, 9). However, USP1–UAF1 targets multiple substrates, including the inhibitor of DNA-binding proteins and TBK1, meaning a catalytic activity inhibitor may affect multiple processes. It is possible that a more specific inhibitor could block FANCD2–USP1 interaction, such as the N-terminus, and may provide a more specific therapy.

# Materials and Methods

### Cloning

Complementary DNA for human full-length USP1$^{FL}$ with G670A/G671A (mutated autocleavage site (7)) was ligated into a pFAST-BAC vector with a TEV cleavable His tag by BamH-Not1 restriction cloning. Subsequently, truncations of USP1 were made in the full-length background by PCR mutagenesis: N-terminus (Δ1–54), Insertion 1 (Δ229–408), and Insertion 2 (Δ608–737). UAF1 was ligated into a pFASTBAC vector with an N-terminal His-3C tag using BamH1-Not1. Mutagenesis was used to add a Strep11 tag and a linker "GGSGGS" N-terminal of the His-tag. X. laevis FANCD2 and X. laevis FANCI genes were a kind gift from Puck Knipscheer (Hubrecht Institute, Utrecht); the genes were ligated into pFASTBAC vectors using EcoRI-XhoI and mutated to contain N-terminal His-TEV-HA and His-TEV-V5 tags, respectively. Human FANCD2 gene was synthesised by GeneArt Gene Synthesis (Invitrogen), expression optimised for

Spodoptera frugiperda, and subsequently cloned into a pFASTBAC vector with a His-3C tag using BamH1-Not1.

USP1$^{Δ1Δ2}$ gene expression optimised for E. coli was synthesised by GeneArt Gene Synthesis (Invitrogen) and subsequently ligated into a pGEX6bp vector containing GST-3C tag by using BamH1-Not1. USP1 amino acids 1–60 (N-terminal) were amplified with BamH1 at either end for ligation into a pGex6bp1 vector containing the rat USP2 catalytic domain (GST-3C-USP1$^{1–60}$-rUSP2 (271–618)). A mutant Ube2T with E54R, P93G, P94G, 1–152 (Ube2Tv4) is described in Chaugule et al, (33). The FANCL$^{ΔELF}$ (109–375) gene was cloned into a pET-SUMO (Invitrogen) vector using restriction-free cloning. Tagless ubiquitin, His-TEV ubiquitin, and His-TEV-PCNA are previously described (22, 48). RNF4 (RNF4-RING domain fusion (34)) was a kind gift from Ron Hay (University of Dundee). To introduce point mutations or truncations, PCR mutagenesis was performed using KOD Hot Start DNA polymerase by following the manufacturer's instructions.

### Protein expression and purification

Baculovirus were generated using the EMBacY MultiBac system (Geneva Biotech), and USP1, UAF1, FANCD2, and FANCI proteins were expressed for ~72 h in baculovirus-infected Sf21 cells. FANCL$^{ΔELF}$, PCNA, ubiquitin constructs, RNF4$^{RING-RING}$, and E2s were expressed and purified in BL21 E. coli, as previously described (34, 48). USP1$^{Δ1Δ2}$ and USP2 were expressed using BL21 E. coli, grown to OD$_{600}$: 0.6 and induced with 0.1 μM IPTG for 18 h at 16°C. Fluorescently labelled ubiquitin was made as described previously (42).

All steps of purifications were performed at 4°C and completed within 24–36 h of lysis. Cell pellets were routinely suspended in lysis buffer (50 mM Tris, pH 8.0, 150 mM NaCl, 5% glycerol, 10 mM imidazole, 10 mM 2-mercaptoethanol with freshly added MgCl$_2$ [2 mM], protease inhibitor [EDTA-free] tablets, and benzonase). For FANCD2, FANCI, and FANCL$^{ΔELF}$, the lysis buffer contained 400 mM NaCl. Sf21 cells were lysed by a homogeniser, followed by sonication at 40% amplitude, 10 s/10 s off for 12 cycles. Suspended E. coli cells were sonicated at 80% amp, 20 s/40 s off for 12 cycles. All lysates were clarified at 40,000 g for 45 min and filtered (0.45 μM). Proteins were bound to respective resins (either Ni-NTA for His tags or glutathione [GSH] for GST tags) and washed extensively with lysis buffer with 500 mM NaCl. GST-tagged proteins were eluted by incubation with GST-precision protease or lysis buffer with 10 mM GSH. His-Smt3–tagged proteins were eluted by on-column cleavage using His-ULP1 protease. His-TEV– or His-3C–tagged proteins were eluted by lysis buffer with 250 mM imidazole and lower NaCl for ion exchange chromatography (typically 100 mM NaCl). To remove tags, proteins were incubated with respective proteases (His-TEV protease or GST-precision protease) overnight and tagged proteases were removed by binding to respective resins. Tags were not removed from FANCD2, FANCI, USP2, or GST-ubiquitin. Anion exchange for USP1, UAF1, FANCD2, FANCI, GST-USP2, GST-Ub, and PCNA was performed using a high-performance Q (1 or 5 ml) column and the column eluted with a linear gradient (50 mM Tris, pH 8.0, 100–1,000 mM NaCl, 5% glycerol, and 10 mM 2-mercaptoethanol). E. coli–expressed USP1 was not purified by anion exchange.

Proteins were further purified using SEC. FANCD2 and FANCI were fractionated using an S6 Increase (10/300) column in 20 mM Tris, pH

8.0, 400 mM NaCl, 5% glycerol, and 5 mM DTT. USP1, UAF1, and E3s were fractionated using an SD200 Increase (10/300) column in 20 mM Tris, pH 8.0, 150 mM NaCl, and 5% glycerol with 5 mM DTT. All ubiquitin constructs and E2s were fractionated using an SD75 (10/300) column in 20 mM Tris, pH 8.0, 150 mM NaCl, 5% glycerol, and 0.5 mM Tris 2-carboxyethyl-phosphine (TCEP). Proteins were concentrated to ~5 mg/ml for FANCD2/FANCI, ~4 mg/ml for FANCL$^{\Delta ELF}$, and ~10 mg/ml for all other proteins before flash freezing in liquid nitrogen and storage at −80°C.

## Analytical gel filtration of USP1–UAF1 complexes

Recombinant USP1 and UAF1 alone or together were incubated at 15 $\mu$M for 15 min on ice before fractionation on an SD200 Increase (10/300) column in 20 mM Tris, pH 8.0, 150 mM NaCl, 5% glycerol, and 5 mM DTT. Samples (10 $\mu$l) were mixed with SDS–PAGE loading buffer (10 $\mu$l) and analysed via SDS–PAGE (9%) and Instant Blue Coomassie staining (Expedeon). All USP1 constructs were co-eluted with UAF1.

## Ub-prg

BL21 *E. coli* carrying Ub-intein-CBD was grown to OD$_{600}$: 0.6, induced with 0.5 $\mu$M IPTG and expressed for 24 h at 16°C. Cells were harvested and suspended in 50 mM Na$_2$HPO$_4$, pH 7.2, 200 mM NaCl, and 1 mM EDTA before lysis and clarification. Lysates were bound to chitin resin and washed extensively in 50 mM Na$_2$HPO$_4$, pH 7.2, and 200 mM NaCl. The resin was washed with wash buffer (20 mM Na$_2$HPO$_4$, pH 6.0, 200 mM NaCl, and 0.1 mM EDTA) before incubation overnight at 4°C with wash buffer containing 100 mM MESNa. The eluent was collected and a second elution was performed in wash buffer plus 100 mM MESNa. The ubiquitin-MESNa was concentrated to ~5 ml and buffer exchanged into 50 mM Hepes, pH 8.0. Ubiquitin-MESNA was either stored at −80°C or directly reacted with 0.25 M propargylamine (prg) for 4 h at 20°C with mild shaking in the dark. Ub-prg was finally purified using SD75 (16/60) in 50 mM Hepes, pH 8.0, and 150 mM NaCl, and flash frozen for storage at −80°C.

## Reacting Ub-prg with DUBs

To crosslink DUBs with Ub-prg, DUBs (2 $\mu$M) in 20 $\mu$l were incubated in DUB buffer (50 mM Tris, pH 7.5, 120 mM NaCl, and 10 mM DTT) with and without Ub-prg (6 $\mu$M) for 10 min at room temperature. Reactions were stopped by the addition of SDS–PAGE loading buffer (Invitrogen) and visualised using 4–12% Bis-Tris SDS–PAGE (Invitrogen) run with NuPAGE MOPs SDS running buffer (Thermo Fisher Scientific) and Coomassie staining. A slower migrating band indicates reaction with Ub-prg.

## Thermofluor assays

Thermofluor experiments were carried out using a CFX96 real-time PCR detection system (Bio-Rad). Samples containing USP1 (40 $\mu$l) at 0.05–0.2 mg/ml in 50 mM Tris, pH 8.0, 120 mM NaCl, 10 mM DTT, and 5× SYPRO orange were dispensed in 96-well plates. The samples were heated from 25°C to 95°C with increments of 1°C/min, and fluorescence was measured at each interval. Data were analysed as

previously described (49), and a mean T$_m$°C was calculated from three different USP1 concentrations performed in triplicate.

## Fluorescent polarisation (FP) assays with ubiquitin-TAMRA assays

USP1–UAF1 samples were prepared by diluting USP1 variants (10 $\mu$M) and UAF1 (10 $\mu$M) in DUB buffer for 10 min at room temperature. For FP assays, USP1–UAF1 samples were diluted and incubated in DUB buffer with 0.1 mg/ml ovalbumin at 2× concentration indicated in reaction. Typically, reactions with Ub-KG$^{TAMRA}$ (UbiQ) were started by adding and mixing 10 $\mu$l of 2× DUB (20 nM) to 10 $\mu$l 2× Ub-KG$^{TAMRA}$ (600 nM) to make a final concentration of 300 nM substrate with 10 nM DUB with a volume of 20 $\mu$l. Reactions were monitored at 25°C for 1 h by measuring FP at 2-min intervals in 384-well round-bottom corning black plates with a PHERAstar FSX. FP values for each well were fitted using a "one-phase decay" model in Prism (GraphPad). pMax was monitored using Ub-KG$^{TAMRA}$ without DUB and pMin using KG$^{TAMRA}$.

## Purifying monoubiquitinated proteins

Purification of PCNA monoubiquitinated at K164 (PCNA-Ub) was performed as previously described (25). Monoubiquitinated FANCD2 at K561 and FANCI at K523 were purified using GST-Ub, which leaves a "GPLGS" over hang at the N-terminus of ubiquitin following precision protease cleavage. Typically, purified FANCD2 or FANCI (after anion exchange) at 4 $\mu$M was monoubiquitinated by equimolar Ube2Tv4, FANCL$^{\Delta ELF}$, 50 nM E1, and 8 $\mu$M GST-3C-ubiquitin in E3 buffer (50 mM Tris, pH 8.5, 150 mM NaCl, 5% glycerol, 1 mM TCEP, 2.5 mM MgCl$_2$, and 2.5 mM ATP) for 30 min at room temperature. *xl*FANCD2 ubiquitination was performed for 10 min rather than 30 min. Reactions were arrested using apyrase followed by anion exchange chromatography (high-performance Q 1 ml). Importantly, prolonged incubation of FANCD2 in low NaCl concentrations (below ~200 mM NaCl) will lead to large losses in protein. GST-Ub-substrates were bound to GSH resin at 4°C before extensive washes using 50 mM Tris, pH 8.0, 400 mM NaCl, 5% glycerol, and 2 mM DTT. Ubiquitinated proteins were eluted from GSH resin using GST-precision protease. Finally, ubiquitinated substrates were purified via gel filtration using an S6 Increase (10/300) column in 20 mM Tris, pH 8.0, 400 mM NaCl, 5% glycerol, and 5 mM DTT before concentrating to ~5 mg/ml and flash freezing in liquid nitrogen for storage at −80°C. Ubiquitinated FANCD2 or FANCI proteins were visualised on Bis-Tris 4–12% gradient SDS–PAGE gels run in MOPs SDS running buffer, the gels were "overrun" so the 20-kD reference band in the Precision Plus prestained protein ladder (Bio-Rad) was at the bottom of the SDS–PAGE gel. To purify FANCD2-Ub$^{800}$, FANCD2 was monoubiquitinated with ubiquitin$^{800}$ instead of GST-Ub, and subsequently purified by anion exchange and gel filtration as above.

## Mass spectrometry

Recombinant FANCD2, FANCI, FANCD2-Ub, and FANCI-Ub (10 $\mu$g) were denatured in 7 M urea and 100 mM Tris, pH 8.0, buffer followed by treatment with 5 mM TCEP (10 min), 5 mM idoacetamide (20 min), and 5 mM DTT (20 min). Samples were diluted 1:10 in 100 mM Tris, pH

8.0, and trypsin was added at 1:50 wt/wt with incubation at 37°C overnight; finally, 1% TFA was added. The peptides were subjected to mass spectrometric analysis performed by LC-MS-MS on a linear ion trap-Orbitrap hybrid mass spectrometer (Orbitrap-VelosPro; Thermo Fisher Scientific) coupled to a U3000 RSLC HPLC (Thermo Fisher Scientific). Data files were analysed by Proteome Discoverer 2.0 (www.ThermoScientific.com), using Mascot 2.4.1 protein identification software (www.matrixscience.com) by searching against an in-house protein database.

## Deubiquitination reactions using recombinant substrates

USP1–UAF1 complexes were assembled and used to make 2× DUB stocks at 200 nM, or as indicated in Fig 3C. Ubiquitinated substrates were made as 2× stocks: purified *hs*FANCD2-Ub (2 $\mu$M), *hs*FANCI-Ub (2 $\mu$M), *hs*PCNA-Ub (6 $\mu$M), *xl*FANCD2-Ub (2 $\mu$M), and diubiquitin chains (20 $\mu$M) were incubated in DUB buffer on ice for 10 min. To initiate hydrolysis, DUBs (200 nM) and substrates (2 or 6 $\mu$M) were incubated in a 1:1 ratio (typically 5 $\mu$l: 5 $\mu$l) for 30 min at room temperature or an indicated time. Reactions were terminated using SDS–PAGE loading buffer (Invitrogen). To analyse deubiquitination, ~300 ng of FANCD2-Ub or FANCI-Ub, ~600 ng of PCNA-Ub, or 800 ng of diubiquitin was separated on 4–12% Bis-Tris SDS–PAGE gels and Coomassie stained. Deubiquitination was determined by the loss of ubiquitinated proteins or the emergence of non-ubiquitinated substrates and monoubiquitin. For USP2 DUB assays, the intensity of ubiquitinated substrates was measured using a 700-nm channel after Coomassie staining, calculated as mean % of the input and plotted against time, with error bars showing standard deviation from the mean.

## Ubiquitination reactions

FANCD2 at K561 and FANCI at K523 were ubiquitinated at room temperature in E3 reaction buffer using Ube2Tv4 and FANCL$^{\Delta ELF}$. For small-scale reactions (10–100 $\mu$l), protein concentrations were typically 2 $\mu$M substrate/E2/E3, 4 $\mu$M ubiquitin, and 50 nM E1. To ubiquitinate FANCD2 at alternative lysines to K561 (i.e., FANCD2-KX-Ub), FANCD2 K561R was ubiquitinated using 50 nM E1, 4 $\mu$M ubiquitin$^{800}$, 0.5 $\mu$M UbcH5c$^{S22R}$, and 0.2 $\mu$M RNF4$^{RR}$ in E3 reaction buffer at room temperature. FANCD2–FANCI–dsDNA complexes were mixed in a ratio of 1:1:2 on ice in 50 mM Tris, pH 7.5, 100 mM NaCl, 5% glycerol, and 1 mM DTT for at least 30 min before being ubiquitinated for 30 min at room temperature using tagless Ub. The dsDNA oligo used is an 84-nucleotide duplex made from a combination of oligos 10 and 11 from Longerich et al (36). Reactions were stopped by the addition of 3× SDS–PAGE loading buffer, or if used for subsequent DUB reactions, the sample was treated with apyrase for 5 min on ice. Ubiquitination was analysed via 4–12% Bis-Tris SDS–PAGE and Odyssey licor CLx with an 800-nm channel for ubiquitin$^{800}$, Western blotting, and/or Coomassie staining.

## DUB-step reactions

DUB-step reactions followed three steps: (1) ubiquitination (see the Ubiquitination reactions section), (2) arrest with apyrase to stop ubiquitination, and (3) deubiquitination. Briefly, FANCD2 (2 $\mu$M)

ubiquitinated on K561 or KX was immediately diluted 5 $\mu$l: 5 $\mu$l with DUBs (200 nM) diluted in DUB buffer, giving a final concentration of DUB at 100 nM and FANCD2 at 1 $\mu$M. DUB-steps were left at room temperature for an indicated duration (usually 30 min), and 300 ng of FANCD2-Ub$^{800}$ was visualised on SDS–PAGE (4–12% Bis-Tris) with an 800-nM channel followed by Coomassie staining. The percentage of residual FANCD2-Ub$^{800}$ was calculated as a percentage of the input and plotted for each DUB. For non-fluorescent FANCD2–Ub–FANCI–DNA complexes, 300 ng of FANCD2 was loaded on 4–12% Bis-Tris SDS–PAGE for Coomassie staining and 50 ng for Western blotting.

## Pull-down assays

N-terminal Strep-tagged UAF1 was complexed with *E. coli*–expressed USP1$^{\Delta 1\Delta 2}$ or USP1$^{\Delta N\Delta 1\Delta 2}$. Subsequently, 10 $\mu$g of USP1–UAF1 was bound to 10 $\mu$l MagStrep-avidin beads equilibrated in pull-down buffer (50 mM Tris, pH 7.5, 120 mM NaCl, 5% glycerol, 1 mM DTT, 0.01% Triston X-100, and 0.01% ovalbumin) for 1 h on ice. Unbound USP1–UAF1 complexes were washed from beads with 3× 60 $\mu$l washes. USP1–UAF1 beads were then divided into separate tubes (10 $\mu$l beads + 10 $\mu$l buffer) and 10 $\mu$g of recombinant substrate was added and incubated for 1 h on ice with gentle agitation. The supernatant was removed and the beads were washed 3× with 40 $\mu$l buffer. Protein was eluted by suspending the beads in 20 $\mu$l pull-down buffer and 10 $\mu$l 3× SDS–PAGE loading buffer and boiling for 2 min. Samples (10 $\mu$l) were run on 4–12% Bis-Tris SDS–PAGE gels. Beads without USP1–UAF1 were also incubated with the substrate to control for nonspecific binding to beads.

## Western blots and antibodies

SDS–PAGE-separated proteins were transferred to nitrocellulose membranes and blocked with 5% milk TBS-T (0.05% tween) before incubation with 1:1,000 anti-FANCD2 (Ab2187; Abcam), 1: 5,000 anti-V5 (66007.1-Ig; ProteinTech), or 1: 2,000 anti-PCNA (Ab29; Abcam) overnight at 4°C. Membranes were washed 3× with TBS-T before incubation with secondary antibodies for 1 h at room temperature and washed with TBS-T. Results were visualised on licor using an 800- or 700-nm channel.

# Supplementary Information

# Acknowledgements

We thank Axel Knebel (University of Dundee) for His-Ube1, and Robert Gourlay from the Medical Research Council Protein Phosphorylation and Ubiquitylation Unit Proteomics Facility (University of Dundee) for mass spectrometry. This work was supported by the Medical Research Council (grant MC_UU_12016/12); the European Molecular Biology Organization Young Investigator Programme to H Walden; and the European Research Council consolidator grant (ERC-2015-CoG-681582 ICLUb) to H Walden.

## Author Contributions

C Arkinson: conceptualization, data curation, formal analysis, investigation, methodology, and writing—original draft, review, and editing.
VK Chaugule: conceptualization, resources, formal analysis, methodology, and writing—review and editing.
R Toth: resources.
H Walden: conceptualization, supervision, funding acquisition, and writing—review and editing.

## Conflict of Interest Statement

The authors declare that they have no conflict of interest.

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
