## [Reviewer comments · Life Science Alliance]

Life Science Alliance

Specificity for deubiquitination of monoubiquitinated FANCD2 is driven by the N-terminus of USP1

Helen Walden, Connor Arkinson, Viduth Chaugule, and Rachel Toth
DOI: 10.26508/lsa.201800162

Corresponding author(s): Helen Walden, RI Mol, Cell, Systems Biol

Review Timeline:

Submission Date:	2018-08-15
Editorial Decision:	2018-08-16
Revision Received:	2018-10-01
Editorial Decision:	2018-10-02
Revision Received:	2018-10-03
Accepted:	2018-10-04

Scientific Editor: Andrea Leibfried

Transaction Report:

Please note that the manuscript was previously reviewed at another journal and the reports were taken into account in inviting a revision for publication at *Life Science Alliance* prior to submission to *Life Science Alliance*.

REFEREE REPORTS OBTAINED DURING PEER REVIEW ELSEWHERE

Referee #1 Review

Report for Author:

The manuscript by Arkinson et al describes a biochemical reconstitution of USP1 deubiquitination of monoubiquitinated FANCD2 and the mapping of critical residues on USP1 that determines substrate specificity of USP1 for FANCD2. The study is sorely needed to clarify some misconceptions of what drives USP1 substrate specificity, especially for the three most well-characterized substrates, FANCD2, FANCI, and PCNA. The study feels like a prelude to a structural determination of USP1-FANCD2 interaction, but given that all of their analysis is done using in vitro assays, it falls short of what is typical for publication at this journal.

Here are some major points that could be helpful for the authors (not in order of importance):

- 1) Although data showing the molecular determinants for substrate deubiquitination (K561 monoubiquitin-specific FANCD2) by USP1 reside within the highly conserved and extended N-terminus is strong, the experimental conditions the authors use are not exhaustive. For instance, are there any noticeable differences between the activities of USP1 (intrinsic catalytic activity vs di-Ub cleavage vs FANCD2 or PCNA deubiquitination) when comparing recombinant USP1 prepared from Sf21 or bacterial cells? What about if USP1 complex was partially purified from human cells? Perhaps some of the discrepancy in the requirement of different domains on WDR48 (UAF1) or on USP1 for substrate specificity is based on post-translational modifications during the prep of the proteins.
- 2) The generation and purification of K561-specific monoubiquitinated hFANCD2 is quite challenging and the authors have done an admirable job with the generation of different recombinant monoubiquitinated proteins, including FANCD2, FANCI and PCNA. Have the authors considered putting both FANCD2 and FANCI together for their USP1 binding studies or for their deubiquitination? Perhaps the N-terminal specificity of USP1 for substrate recognition is only retained when both FANCD2 and FANCI are bound together. Assessing USP1 substrate specificity on FANCD2 and FANCI individually may not be representative of their in vivo states. There are lots of in vitro and in vivo evidence that FANCD2 and FANCI interact together in the unmodified and/or monoubiquitinated state.
- 3) Similarly, monoubiquitinated PCNA occurs on DNA primer-template junction with Pol delta/epsilon. Would this configuration facilitate N-terminal specificity of USP1 for substrate recognition and deubiquitination?
- 4) The USP1 wildtype and mutants (delta1 vs delta2 vs delta 1+2 vs delta N vs delta N+1+2, vss R22A vs R22E vs R22K) need to be all individually reconstituted in a USP1-deficient cell line to determine whether their in vitro functions can be truly recapitulated in cell-based assays (western blots and cell survival studies). It will be interesting to see what effect of the R22K would have on differential deubiquitination of FANCD2 vs FANCI vs PCNA. In previous studies, the deubiquitination of FANCD2 vs FANCI have never been uncoupled (Ub-FANCD2 high while Ub-FANCI is low). It will be interesting to see if this state can support HDR-type DNA repair.
- 5) The results in Figure 6B is quite difficult to interpret. Why would an increase in DUB

concentration of USP2 be able to deubiquitinate FANCD2, but an increase in DUB concentration of USP1 (deltaN+1+2) could not? It appears that USP1 has additional determinants that prevent USP1 delta from deubiquitinating FANCD2 at the very high concentration. USP2 and USP1 intrinsic catalytic activities are likely different. In order to conclusively show that the USP1 N-terminal domain can act as a substrate specificity factor in trans for USP2 as a chimera protein, USP1 and USP2 DUB activities against different substrates need to be carefully assessed. Time course and dose experiments need to be shown and done in triplicates and graphed with statistical significance. The better experiment would be to use another DUB that also requires WDR48 (UAF1) for activity and to determine whether adding the USP1 N-terminal domain can convert USP12/46 to a FANCD2 deubiquitinase. USP2's intrinsic DUB activity is likely too robust and non-selective.

Referee #2 Review

Report for Author:

This manuscript reports the identification of a specific interaction face between the N-terminal region of USP1 and FANCD2 required for the de-ubiquitination of FANCD2 K561. It is intriguing the catalytic domain of USP1 with its partner UAF1 is sufficient for generic DUB activity but not for specific FANCD2 de-ubiquitination, which requires the N-terminus. Moreover the N-terminus is sufficient for this specificity and a chimera bearing it can direct a non-specific DUB to FANCD2.

Significant technical obstacle overcome to make this study possible: generation of Mono-Ub FANCD2 is not trivial.

Several surprising omissions: that point mutations in the N-terminus that exhibit reduced DUB activity were not tested for FANCD2 interaction and then used in the chimera experiment.

Not clear whether targeting of USP1 to FANCD2 also enables FANCI deubiquitination in the context of the heterodimer.

Current paper is of high technical quality, but other DUBs have been identified that have substrate specificity - albeit many not yet mapped to identify the structural determinant. Here that is not complete since the nature of the N-terminal requirement not entirely clear - correlation between DUB activity and interaction not made and the reciprocal site not mapped.

More significantly there is no cellular assessment (physiological significance) to show that the N-terminus and particularly the point mutations identified have any role in the cellular FA response.

Referee #3 Review

Report for Author:

This manuscript addresses the basis for the distinct substrate specificity of USP1, which is one of several USP deubiquitinating enzymes that function as a heterodimer with UAF1 that targets substrates in the Fanconi Anemia Pathway. USP1/UAF1 deubiquitinates FANCD2, FANCI and PCNA. In this study, the authors assay the effects of a variety of deletions and point mutations on both the activity and specificity of USP1/UAF1 for these three substrates and find that the N-terminus of USP1 is specifically required for the ability of USP1 to remove monoubiquitin from FANCD2-K561. Using a clever approach in which they mutate K561 and then ubiquitinate other lysines in FANCD2, they show that the role of the N-

terminus is very specific to targeting USP1 to K561 and not to other ubiquitinated lysines in the substrate. They map the targeting sequence to a short N-terminal peptide using alanine mutations. The authors show that the N-terminal 60 residues are sufficient to confer specificity for FANCD2, as they show that a chimeric protein containing the USP1 N-terminus spliced onto USP2 can confer on USP2 the ability to deubiquitinate FANCD2.

The conclusions of the manuscript, which are largely well-supported by the data, add to an understanding of the substrate specificity determinants of USP1 but do not represent a sufficient advance for the audience of this Journal. There is no insight here as to how the USP1 confers specificity for FANCD2, which could perhaps be provided with more mechanistic studies and better characterization of the effects of deletions and mutations on USP1 kinetic parameters or quantitative measures of substrate binding. Some sort of *in vivo* result showing the importance of the USP1 N-terminus in cells could also bolster the case for the importance of interactions with the N-terminus in either directing USP1 to bind to specific substrates or modulating its activity in a substrate-dependent manner.

Specific major points:

- 1) The conclusion that ubiquitin provides additional binding energy in the case of the USP1 Δ N Δ 1 Δ 2 mutant is not clearly supported by the pull-downs in figure 6A. First, it was not clear what the lanes labeled 1 2 3 4 were (are these washes? Time points?). More importantly, the hFANCD2-Ub band for that mutant is extremely faint, raising the question of how different that binding is than that to the unmodified FANCD2, which arguably may fall just below the level of detection. While it is certainly plausible that the conjugated ubiquitin can confer additional binding energy, the data shown are not sufficiently definitive.
- 2) The statement in the discussion that deletion of the USP1 N-terminus disrupts the ability of the enzyme to form a productive enzyme-substrate complex could be tested by assaying the effects of the deletion on *k_{cat}* and *K_m*. Single-turnover experiments, for example, could provide information on *k_{cat}* without having to generate large quantities of substrate.
- 3) The discussion of USP1 activity in terms of the enzyme:substrate stoichiometry is misleading. This leaves the reader with the impression that it is the ratio of the two that is important, whereas it is the enzyme concentration that matters, either because it increases the rate or because it drives binding to substrate. This could be sorted out with more quantitative studies of the enzymes.

Minor points:

- 1) The times in the end-point assays should be clearly indicated in each figure.
- 2) SUMO should be all caps.
- 3) The * should be identified in Fig. EV4.
- 4) Zebrafish is one word.
- 5) The icon for FANCD2 looks distractingly like a cucumber. Perhaps a slightly different shape could be considered.

1st Editorial Decision August 16, 2018

August 16, 2018

Re: Life Science Alliance manuscript #LSA-2018-00162-T

Prof. Helen Walden
RI Mol, Cell, Systems Biol
University of Glasgow
University avenue
Glasgow G12 8QQ
United Kingdom

Dear Dr. Walden,

Thank you for transferring your manuscript entitled "Specificity for deubiquitination of monoubiquitinated FANCD2 is driven by the N-terminus of USP1" to Life Science Alliance. The manuscript was assessed by expert reviewers at another journal before, and the editor transferred these reports to us.

The reviewers who assessed your work at the other journal before think that your work is important and well performed, but not providing a deep structural and mechanistic understanding. The latter is not a concern for publication in Life Science Alliance, and we would like to invite you to submit a slightly revised version for publication here. Please provide a point-by-point response to the criticisms raised and accordingly text changes. If you have data at hand that already address specific concerns raised (such as UAF1 N-terminus mutant testing for interaction with FANCD2; cell-based assays for UAF1 mutants), it would be good to include them.

-- High-resolution figure, supplementary figure and video files uploaded as individual files: See our detailed guidelines for preparing your production-ready images, <http://life-science-alliance.org/authorguide>

B. MANUSCRIPT ORGANIZATION AND FORMATTING:

Full guidelines are available on our Instructions for Authors page, <http://life-science-alliance.org/authorguide>

*****IMPORTANT:** It is Life Science Alliance policy that if requested, original data images must be made available. Failure to provide original images upon request will result in unavoidable delays in publication. Please ensure that you have access to all original microscopy and blot data images before submitting your revision. *******

Thank you for this interesting contribution to Life Science Alliance. We are looking forward to receiving your revised manuscript.

Sincerely,

Andrea Leibfried, PhD
Executive Editor
Life Science Alliance
Meyershofstr. 1
69117 Heidelberg, Germany
t +49 6221 8891 502
e a.leibfried@life-science-alliance.org
www.life-science-alliance.org

1st Authors' Response to Reviewers: October 1, 2018

We thank all three reviewers for taking the time to review our manuscript and for their constructive criticisms. Our response to the comments is below, in bold.

Referee #1:

The manuscript by Arkinson et al describes a biochemical reconstitution of USP1 deubiquitination of monoubiquitinated FANCD2 and the mapping of critical residues on USP1 that determines substrate specificity of USP1 for FANCD2. The study is sorely needed to clarify some misconceptions of what drives USP1 substrate specificity, especially for the three most well-characterized substrates, FANCD2, FANCI, and PCNA. The study feels like a prelude to a structural determination of USP1-FANCD2 interaction, but given that all of their analysis is done using in vitro assays, it falls short of what is typical for publication at this journal.

We are grateful to reviewer 1 for recognising that this study is sorely needed.

Here are some major points that could be helpful for the authors (not in order of importance):

1) Although data showing the molecular determinants for substrate deubiquitination (K561 monoubiquitin-specific FANCD2) by USP1 reside within the highly conserved and extended N-terminus is strong, the experimental conditions the authors use are not exhaustive. For instance, are there any noticeable differences between the activities of USP1 (intrinsic catalytic activity vs di-Ub cleavage vs FANCD2 or PCNA deubiquitination) when comparing recombinant USP1 prepared from Sf21 or bacterial cells? What about if USP1 complex was partially purified from human cells? Perhaps some of the discrepancy in the requirement of different domains on WDR48 (UAF1) or on USP1 for substrate specificity is based on post-translational modifications during the prep of the proteins.

We tested the USP1 constructs we made in insect cells and compared to constructs we made in *E.coli*. In all cases, all constructs react with the Ub-prg with no noticeable differences (these are in figures 1c, and Supp figure 4). We also observe no obvious differences in Sf21 versus *E.coli*-derived USP1 material in the case of FANCD2. However, in the case of monoubiquitinated PCNA, Sf21-produced USP1 appears to remove more Ub from PCNA-Ub than DeltaN-USP1 (already shown in figure 3A). We now compare this material with the *E.coli* material, and find that *E.coli* expressed USP1 or deltaN-USP1, and Sf21-expressed deltaN-USP1 are all similar, while Sf21-FLUSP1 is more efficient (Shown in Supp fig 3). The fact that the *E.coli*-derived material does not depend on the presence of the N-terminus for PCNA-Ub deubiquitination suggests that any differences observed in the Sf21-derived USP1 fragments are likely due to some other unknown regulation, possibly phosphorylation.

Intriguingly though, the difference between Full length and deltaN-USP1 from Sf21 against PCNA-Ub suggests that any regulation that does exist is in the N-terminal motif, which we have identified as important for substrate targeting.

We feel that a study into the *regulation* of USP1 substrate targeting is beyond the scope of the current manuscript. As the reviewer acknowledges, we are reporting the discovery of a substrate-targeting element in USP1, which is made possible by our *in vitro* reconstitution of the minimal components. Determining how this (and potentially other) element(s) are regulated in cells is of course important and interesting, and will form the basis of our follow-up work.

2) The generation and purification of K561-specific monoubiquitinated hFANCD2 is quite challenging and the authors have done an admirable job with the generation of different recombinant monoubiquitinated proteins, including FANCD2, FANCI and PCNA. Have the authors considered putting both FANCD2 and FANCI together for their USP1 binding studies or for their deubiquitination? Perhaps the N-terminal specificity of USP1 for substrate recognition is only retained when both FANCD2 and FANCI are bound together. Assessing USP1 substrate specificity on FANCD2 and FANCI individually may not be representative of their *in vivo* states. There are lots of *in vitro* and *in vivo* evidence that FANCD2 and FANCI interact together in the unmodified and/or monoubiquitinated state.

We are grateful that the reviewer understands the challenge in reconstituting the system *in vitro*. So far, available evidence suggests that FANCD2 and FANCI are only monoubiquitinated when together, and in the presence of DNA (e.g. Sato et al., 2012; Rajendra et al., 2014; van Twest et al., 2017). However, to our knowledge no group has yet examined, in either a purified system or in cells, how the *ubiquitinated* proteins associate – no doubt due to the challenges of preparing the materials. We consider the question of what happens to the proteins and complex when they are ubiquitinated to be interesting and important, but is not the question we are addressing in this study. That being said, we accept the possibility, raised by the reviewer, that investigating USP1 activity with each individual modified protein may not represent the *in vivo* states. We do not know what the *in vivo* states are. To our knowledge, neither does anyone else. However, on the reviewer's suggestion we have performed the following experiments:

Purified FANCI and FANCD2 with DNA added are incubated with FANCL-Ube2T to monoubiquitinate FANCI-FANCD2 in the heterodimeric form. The reaction is arrested, and then treated with USP1 versions. A recent study (van Twest et al., 2017) suggests that USP1 is unable to deubiquitinate FANCI-Ub or FANCD2-Ub in the presence of DNA, therefore we performed the deubiquitination step in the presence or absence of benzonase.

FANCD2 is not fully ubiquitinated in this setup, however it is partially deubiquitinated by FL USP1 (Figure 7A, lane 2). In contrast addition of deltaN or

R22E-USP1 (lanes 3, 4) resemble the no DUB control (lane 1). In the presence of benzonase, USP1 completely deubiquitinates FANCD2-Ub, while deltaN or R22E-USP1 deubiquitinate to a lesser extent – this is in agreement with the recent model from van Twest et al., showing FANCD2-Ub must be removed from DNA in order to be deubiquitinated.

However, van Twest et al., also find that FANCI is needed for FANCD2-Ub deubiquitination. We already observed in fig 3A that this is not the case in our setup.

Therefore, we repeated the heterodimer experiment with the ubiquitination site on FANCI mutated, in order to test whether ubiquitination status of FANCI in the heterodimer impacts FANCD2-Ub deubiquitination. We find that when FANCI is unmodified, FANCD2 is fully deubiquitinated in presence or absence of benzonase by FL-USP1, but NOT by deltaN, or R22E-USP1. These data show that even in the currently proposed “physiological” state, FANCD2-Ub deubiquitination is dependent on the N-terminus of USP1, consistent with our data in the single protein experiments.

We have added these new data as new Figure 7.

3) Similarly, monoubiquitinated PCNA occurs on DNA primer-template junction with Pol delta/epsilon. Would this configuration facilitate N-terminal specificity of USP1 for substrate recognition and deubiquitination?

This is an interesting question which we had not considered. We have added a sentence in discussion to acknowledge this possibility.

4) The USP1 wildtype and mutants (delta1 vs delta2 vs delta 1+2 vs delta N vs delta N+1+2, vs R22A vs R22E vs R22K) need to be all individually reconstituted in a USP1-deficient cell line to determine whether their *in vitro* functions can be truly recapitulated in cell-based assays (western blots and cell survival studies). It will be interesting to see what effect of the R22K would have on differential deubiquitination of FANCD2 vs FANCI vs PCNA. In previous studies, the deubiquitination of FANCD2 vs FANCI have never been uncoupled (Ub-FANCD2 high while Ub-FANCI is low). It will be interesting to see if this state can support HDR-type DNA repair.

The referee raises important points that we are currently addressing. However, we feel an in-depth cellular study to uncouple different states of FANCD2 and FANCI ubiquitination, and what impact this has in HDR-type repair is beyond the scope of the current manuscript.

5) The results in Figure 6B is quite difficult to interpret. Why would an increase in DUB concentration of USP2 be able to deubiquitinate FANCD2, but an increase in DUB concentration of USP1 (deltaN+1+2) could not? It appears that USP1 has additional determinants that prevent USP1 delta from deubiquitinating FANCD2 at the very high concentration. USP2 and USP1 intrinsic catalytic activities are likely different. In order to

conclusively show that the USP1 N-terminal domain can act as a substrate specificity factor in trans for USP2 as a chimera protein, USP1 and USP2 DUB activities against different substrates need to be carefully assessed. Time course and dose experiments need to be shown and done in triplicates and graphed with statistical significance. The better experiment would be to use another DUB that also requires WDR48 (UAF1) for activity and to determine whether adding the USP1 N-terminal domain can convert USP12/46 to a FANCD2 deubiquitinase. USP2's intrinsic DUB activity is likely too robust and non-selective.

Figure 3C shows that if the concentration of deltaN is increased to equimolar with FANCD2-Ub, more ubiquitin is cleaved, we have tried to make that clearer.

We agree that USP2 is robust and non-selective, that is why we chose it as the model for the chimera experiment, which is done in the absence of UAF1. We use USP2 at 100nM (in the presence of 1 micromolar substrate) and show that adding the N-terminus of USP1 augments USP2 to deubiquitinate FANCD2-Ub more efficiently, essentially demonstrating that adding a domain outside of the catalytic domain to another USP enhances its activity towards FANCD2-Ub (but not FANCI-Ub or PCNA-Ub). We appreciate the suggestion for the USP12/USP46 and will try it in our follow up studies.

We have now added the repeats and quantification requested, these are shown in revised figure 6.

Referee #2:

This manuscript reports the identification of a specific interaction face between the N-terminal region of USP1 and FANCD2 required for the de-ubiquitination of FANCD2 K561.

It is intriguing the catalytic domain of USP1 with its partner UAF1 is sufficient for generic DUB activity but not for specific FANCD2 de-ubiquitination, which requires the N-terminus. Moreover the N-terminus is sufficient for this specificity and a chimera bearing it can direct a non-specific DUB to FANCD2.

Significant technical obstacle overcome to make this study possible: generation of Mono-Ub FANCD2 is not trivial.

We are grateful to the reviewer for understanding that we identify a specific interaction module within USP1, and the development of making monoubiquitinated FANCD2.

Several surprising omissions: that point mutations in the N-terminus that exhibit reduced DUB activity were not tested for FANCD2 interaction and then used in the chimera experiment.

We have tested the R22A point mutation and find that it has reduced binding with FANCD2 (supp figure 7A), suggesting that the reduced activity of this point mutant is at least in part due to reduced binding.

Not clear whether targeting of USP1 to FANCD2 also enables FANCI deubiquitination in the context of the heterodimer.

Please see response to reviewer 1, point 2.

Current paper is of high technical quality, but other DUBs have been identified that have substrate specificity - albeit many not yet mapped to identify the structural determinant. Here that is not complete since the nature of the N-terminal requirement not entirely clear- correlation between DUB activity and interaction not made and the reciprocal site not mapped.

More significantly there is no cellular assessment (physiological significance) to show that the N-terminus and particularly the point mutations identified have any role in the cellular FA response.

All of these points are currently under investigation. We have not yet been able to map the reciprocal site, the FANCD2/FANCI human proteins are technically challenging to work with, particularly for more quantitative binding studies. An in-depth cell based analysis of the substrate targeting by USP1 is something we are working on, but we feel is beyond the scope of this manuscript.

Referee #3:

This manuscript addresses the basis for the distinct substrate specificity of USP1, which is one of several USP deubiquitinating enzymes that function as a heterodimer with UAF1 that targets substrates in the Fanconi Anemia Pathway. USP1/UAF1 deubiquitinates FANCD2, FANCI and PCNA. In this study, the authors assay the effects of a variety of deletions and point mutations on both the activity and specificity of USP1/UAF1 for these three substrates and find that the N-terminus of USP1 is specifically required for the ability of USP1 to remove monoubiquitin from FANCD2-K561. Using a clever approach in which they mutate K561 and then ubiquitinate other lysines in FANCD2, they show that the role of the N-terminus is very specific to targeting USP1 to K561 and not to other ubiquitinated lysines in the substrate. They map the targeting sequence to a short N-terminal peptide using alanine mutations. The authors show that the N-terminal 60 residues are sufficient to confer specificity for FANCD2, as they show that a chimeric protein containing the USP1 N-terminus spliced onto USP2 can confer on USP2 the ability to deubiquitinate FANCD2.

We are grateful to this reviewer for their clear understanding of our manuscript, and particularly for highlighting that we addressed the deubiquitination of a specific lysine, not just a specific substrate.

The conclusions of the manuscript, which are largely well-supported by the data, add to an understanding of the substrate specificity determinants of USP1 but do not represent a sufficient advance for the audience of this journal. There is no insight here as to how the USP1 confers specificity for FANCD2, which could perhaps be provided with more mechanistic studies and better characterization of the effects of deletions and mutations on USP1 kinetic parameters or quantitative measures of substrate binding. Some sort of in vivo result showing the importance of the USP1 N-terminus in cells could also bolster the case for the importance of interactions with the N-terminus in either directing USP1 to bind to specific substrates or modulating its activity in a substrate-dependent manner.

We agree with the reviewer that our findings prompt several questions regarding the mechanism of USP1 specificity for FANCD2. This will form the basis of future studies.

Specific major points:

1) The conclusion that ubiquitin provides additional binding energy in the case of the USP1 Δ N Δ 1 Δ 2 mutant is not clearly supported by the pull-downs in figure 6A. First, it was not clear what the lanes labeled 1 2 3 4 were (are these washes? Time points?). More importantly, the hFANCD2-Ub band for that mutant is extremely faint, raising the question of how different that binding is than that to the unmodified FANCD2, which arguably may fall just below the level of detection. While it is certainly plausible that the conjugated ubiquitin can confer additional binding energy, the data shown are not sufficiently definitive.

We agree with the reviewer that a pulldown assay is not sufficient to support the statement that ubiquitin provides additional binding energy. The proteins are on the same membrane and measured for intensity, but that is not quantitative. All this experiment shows is that the N-terminus binds to FANCD2-Ub, and until we've established a more quantitative assay (e.g. ITC) we've decided to move these data to supplementary and tone the statement down to show the N-terminus supports the interaction. For completeness, FANCD2 and FANCD2-Ub are prone to aggregation at the concentrations needed for ITC analysis, and we are still trying to overcome that technical constraint.

2) The statement in the discussion that deletion of the USP1 N-terminus disrupts the ability of the enzyme to form a productive enzyme-substrate complex could be tested by assaying the effects of the deletion on k_{cat} and K_m . Single-turnover experiments, for example, could provide information on k_{cat} without having to generate large quantities of substrate.

The reviewer raises an important point, and one that we have tried to address in a fluorescence polarisation setup, where the ubiquitin is fluorescently labelled and cleavage results in a decrease in polarisation. Our preliminary data suggest that

at high concentration, both FANCD2 and FANCD2-Ub are prone to aggregation, making the polarisation uninterpretable, and therefore experiments are challenging and may be difficult to interrupt. We are still working on developing a better assay for exactly these questions, and have therefore removed this statement from the discussion.

3) The discussion of USP1 activity in terms of the enzyme: substrate stoichiometry is misleading. This leaves the reader with the impression that it is the ration of the two that is important, whereas it is the enzyme concentration that matters, either because it increases the rate or because it drives binding to substrate. This could be sorted out with more quantitative studies of the enzymes.

We have clarified this statement in the discussion, as indeed, we were trying to make the point that it is the enzyme concentration that matters.

Minor points:

- 1) The times in the end-point assays should be clearly indicated in each figure. **Added.**
- 2) SUMO should be all caps. **Changed.**
- 3) The * should be identified in Fig. EV4. **Added.**
- 4) Zebrafish is one word. **Changed.**
- 5) The icon for FANCD2 looks distractingly like a cucumber. Perhaps a slightly different shape could be considered.

The representation we want to use is the shape of the protein rendered in space filling using pymol. Please see below.

October 2, 2018

RE: Life Science Alliance Manuscript #LSA-2018-00162-TR

Prof. Helen Walden
RI Mol, Cell, Systems Biol
University of Glasgow
University avenue
Glasgow G12 8QQ
United Kingdom

Dear Dr. Walden,

Thank you for submitting your revised manuscript entitled "Specificity for deubiquitination of monoubiquitinated FANCD2 is driven by the N-terminus of USP1". I appreciate the introduced changes and am happy to publish your paper in Life Science Alliance pending final revisions necessary to meet our formatting guidelines:

- please provide all figures without figure legends and upload them as individual files (also the Suppl Figures)
- please include 10 authors et al in your reference list
- please note that there is a callout for Fig4D in the ms text, though there is no panel
- please add a callout in the manuscript text to FigS4C and FigS4D

A. FINAL FILES:

-- High-resolution figure, supplementary figure and video files uploaded as individual files: See our detailed guidelines for preparing your production-ready images, <http://life-science-alliance.org/authorguide>

B. MANUSCRIPT ORGANIZATION AND FORMATTING:

Full guidelines are available on our Instructions for Authors page, <http://life-science-alliance.org/authorguide>

Sincerely,

October 4, 2018

RE: Life Science Alliance Manuscript #LSA-2018-00162-TRR

Prof. Helen Walden
RI Mol, Cell, Systems Biol
University of Glasgow
University avenue
Glasgow G12 8QQ
United Kingdom

Dear Dr. Walden,

Thank you for submitting your Research Article entitled "Specificity for deubiquitination of monoubiquitinated FANCD2 is driven by the N-terminus of USP1". It is a pleasure to let you know that your manuscript is now accepted for publication in Life Science Alliance. Congratulations on this interesting work.

The final published version of your manuscript will be deposited by us to PubMed Central (PMC) as soon as we are allowed to do so, the application for PMC indexing has been filed. You may be eligible to also deposit your Life Science Alliance article in PMC or PMC Europe yourself, which will then allow others to find out about your work by Pubmed searches right away. Such author-initiated deposition is possible/mandated for work funded by eg NIH, HHMI, ERC, MRC, Cancer Research UK, Telethon, EMBL.

Please also see:

<https://www.ncbi.nlm.nih.gov/pmc/about/authorms/>
<https://europepmc.org/Help#howsubsmanu>

*****IMPORTANT:** If you will be unreachable at any time, please provide us with the email address of an alternate author. Failure to respond to routine queries may lead to unavoidable delays in publication.*******

DISTRIBUTION OF MATERIALS:

Authors are required to distribute freely any materials used in experiments published in Life Science Alliance. Authors are encouraged to deposit materials used in their studies to the appropriate

repositories for distribution to researchers.

Again, congratulations on a very nice paper. I hope you found the review process to be constructive and are pleased with how the manuscript was handled editorially. We look forward to future exciting submissions from your lab.

Sincerely,
